# Digital Shifts and Ethno-Political Dynamics: Examining Event and Actor Designation in the Cameroon Boko Haram Terrorism Conflict through Print and Online Platforms

Willy Stephane Abondo Ndo 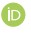

Department of Information and Communication Sciences, Faculty of Arts, Translation, and Communication, Université Libre de Bruxelles, 50-1050 Brussels, Belgium; willy.abondo.ndo@ulb.be

**Abstract:** This study examines how the issue of ethnic identity is approached in Cameroon within the context of combating Boko Haram terrorism, considering the influence of the rise of social media on journalistic practices. The advent of these platforms has fundamentally altered the landscape of media coverage, challenging the traditional monopoly of journalists in shaping the narrative of news. How does this technological shift affect the discourse, especially in the designation of events and actors in the reporting of Boko Haram terrorism in Cameroon, whether in traditional print media or on online platforms like Facebook? Do these designations in print media and Facebook discussion forums indicate shifts in the dynamics of the Cameroonian media sphere (censorship, government repression, etc.), resulting from the emergence of new voices in digital discursive spaces? This study employs a dual analysis, integrating a critical examination of media discourse with a sociological study of journalistic production. The scrutiny of media discourse is based on the investigation of 497 articles published between 1st January and 30 June 2015, sourced from seven Cameroonian newspapers. The online corpus encompasses 450 written publications from three Facebook forums. We aim to establish a dialectical relationship between newspaper discourse, online content, and the sociological foundations shaping their production. The observed quality of designations in the studied forums unveils a surge in hate speech within the ethno-political landscape of Cameroon. While this phenomenon remained manageable through the intervention of state regulatory bodies in traditional media, the unrestrained nature of online content, coupled with the absence of state control, has facilitated the rise of inter-ethnic discursive hatred in politics. In conclusion, this study underscores the challenges stemming from the evolution of journalistic practices in a technological landscape and emphasizes the urgent need for regulatory frameworks to counteract the upswing in hate speech and inter-ethnic tensions within political discourse.

**Keywords:** event and actor designation; Boko Haram terrorism; ethnic identity; print and online platforms; critical analysis of press discourse

## 1. Introduction

On 17 May 2014, on the steps of the Elysée, the Cameroonian president officially declared war against Boko Haram, a terrorist group affiliated with Daesh (the Islamic State). This decision followed numerous attacks by this Islamist sect in the region of the Cameroonian Far North. This area borders the State of Borno in neighboring Nigeria, where this movement originated and established its base. Seven months later, Cameroonian newspapers used various designations to name this event: the rebellion of the North, the northern plot, the insurgency of the Grand North, the specter of 6 April, and the demonstration of strength of the northern front. By a press event, we mean the media treatment of what Alice Krieg-Planque (2009, pp. 77–90) refers to as an occurrence (that is, what happens in the phenomenal world) perceived as significant in a particular context that determines (or not) its remarkability. These designations of the conflict by Cameroonian

newspapers have the particularity of assigning it an identity definition. They associate a specific ethno-regional and religious identity group with this press event, while also invoking the collective memory related to the painful history of the country.

Indeed, the reference to Northern Cameroon in these formulations reflects the fact that Ahmadou Ahidjo, the first president of the country (1960–1982), was Fulani, a Muslim native of this region. On 4 November 1982, he resigned and handed over power to his constitutional successor, Paul Biya, who was sworn in on 6 November 1982, and is still in office.

On 6 April 1984, a faction of the armed forces, predominantly composed of individuals from the region of the first president who remained loyal, attempted a coup d'état. The coup ended in failure after more than 72 h of combat against loyalist and republican forces. Decades later, as the wounds from this failed coup were still healing, Boko Haram attacks on Cameroonian soil were swiftly perceived in public opinion as a resurgence of the reconstituted mutineers seeking revenge. The responsibility of politicians from this part of the country, allegedly supported by France, was implicated (Pommerolle 2015). The Speaker of the Lower House of Parliament (of Fulani and Muslim descent) and the then Minister Delegate for Supreme State Control (belonging to the Béti ethnic conglomerate, like the Head of State) successively denounced Boko Haram's accomplices, mainly in the northern regions and external powers. The tone, language used, the nature of accusations, and reactions in specific newspapers led to framing the coverage of these events within historical conflicts linked to ethno-regional identity trajectories. This phase of the conflict against Boko Haram was labeled by newspapers as the northern conspiracy theory. It triggered several other event sequences, including the war effort of the populations, a national solidarity operation initiated by ethno-political leaders in each region of the country to aid the populations of the Far North, devastated by terrorist attacks, and to support the army on the front line. Support marches and mobilizations on its behalf and motions of support for the Head of State, the Supreme Commander of the Armed Forces, constituted four press event sequences in total.

These narratives and politico-identity clashes extended onto digital social media (DSM). We focused on three Facebook discussion groups: *Le Cameroun c'est le Cameroun* (LCCLC) (Cameroon is Cameroon), *Le Cameroun pour tous* (LCPT) (Cameroon for Everyone) and *Le Cameroun que je veux voir*, (The Cameroon I Want to See). In one of these groups, the Cameroonian army is likened to an ethnic militia, specifically the Bulu militia named after the ethnic group to which the Head of State belongs. Supporters of the ruling party refer to a certain faction of the opposition as Talibans, against the backdrop of the Beti/Bamileke cleavage (the two major ethnic conglomerates around which struggles for the conquest and retention of power crystallize), which marked the last presidential election in October 2018. Our interest in these online discussion forums stems from their prevalence in the Cameroonian political discourse, the high daily publication rates, the substantial number of followers, and the significant interaction among users in these groups. This study focuses on the event and actor designation in both newspapers and online publications within our corpus related to this current affair. It is structured around the research questioning below:

How does this technological shift affect the discourse, especially in the designation of events and actors in the reporting of Boko Haram terrorism in Cameroon, whether in traditional print media or on online platforms like Facebook? Do these designations in print media and Facebook discussion forums indicate shifts in the dynamics of the Cameroonian media sphere (censorship, government repression, etc.), resulting from the emergence of new voices in digital discursive spaces?

In Cameroon, the shift from print media to online content has transformed the relationship with information. The arrival of new actors (webnotes/internet users) means that journalists are no longer the exclusive arbiters of the meaning of news. For Christensen (1997), Andreessen (2011), Tapscott and Tapscott (1994), and Paul Maharg's (2016) terminology, this phenomenon is referred to as disintermediation. This involves a process whereby intermediaries in a supply chain are eliminated, typically through the digital re-engineering

of processes and workflows. This process can lead to the dismantling of nearly all industries and professions and the restructuring of nearly every facet of customer-centric activities. In our context, this pertains to the relationship between journalists and their audiences.

While journalists operated in an environment where their content was regulated by numerous state institutions, new actors operate with more freedom online, especially when outside the country. Locally, new legal frameworks empower authorities to penalize online content creators deemed unlawful, rules that are challenging to enforce on internet users based abroad.

## 2. Theoretical Approach

Our approach revolves around the concept of designation, specifically focusing on event designation and actor designation as forms of informative framing. We delve into how these designations contribute to the unfolding of identity-related phenomena in both newspaper articles and online content within our corpus. In other words, we examine how designative choices shape the perception of the fight against terrorism in Cameroon, establishing the ethno-regional contours and stakes. Amossy (2005) discusses the sociality of discourse in this context, emphasizing that discourse is contingent upon its audience, to which the speaker must adapt (Perelman 1989). Seeking the sociality of the text is, above all, questioning who is speaking to whom, at what moment, from which place, under what circumstances, and in what space (Amossy 2005, p. 61).

Addressing the relationship of designation in this context aligns with the broader semiotic problematics of signs, particularly those that consider the study of meaning and its variations from the perspective of denomination or nomination (Kleiber 1984, 2001, 2003; Frath 2015; Petit 2012), among others. Understanding the reality of the identity dynamics in Cameroon regarding the fight against terrorism through the lens of designation involves considering discursive phenomena that underlie the apprehension of signs by speaking subjects. To do so, it is crucial to distinguish what designation is not from what it is. Designation is not nomination, and certainly not denomination. To illustrate this distinction, we refer to the passage by Kleiber. According to him,

> the two types of relations cannot be confused because, contrary to the sole relationship of designation, the relationship of denomination requires, that the relationship X (linguistic expression)—> x (things) has been established beforehand. Indeed, there is a relationship of denomination between X and x only if and only if there has been a prior act of denomination, i.e., the establishment of a referential link or a referential fixation, which can result from an effective act of denomination or simply from an associative habit, between the element x and the linguistic expression X. Such a requirement is not at all necessary for the relationship of designation. If I can only call something by its name if the thing has been named as such beforehand, I can designate, refer to, or point to something with an expression without that thing having been previously designated as such.

(Kleiber 2001, p. 24)

Denomination is a kind of "label nomenclature, the inventory of which dictionaries compile, capturing the meaning conveyed by discourse" (Longhi 2015, p. 5). Kleiber (2001) attributes several characteristics to it, including a pre-established fixed reference between the linguistic sign and the extralinguistic reality, a durable referential association, a preconstructed representational sense, an existential presupposition, etc. Nomination, on the other hand, categorizes a referent by placing it within a class of objects identified in the lexicon (Longhi 2015, p. 6). According to Frath (2015, p. 39), it is a candidate to become a denomination.

Thus, designation involves referring to an extralinguistic object without any stable and fixed relationship between the linguistic sign and the referent having been previously established. The designator is therefore an unstabilized lexical form, a product of the creative imagination of the speaker. In journalistic work, it serves as a powerful framing tool, directing the perception of recipients through specific designation modalities (Calabrese 2009). This is because journalists select information to disseminate and often

assign designations to the actors they discuss (Calabrese 2009, 2015). In upcoming sections, we will explore how designation operates in newspaper articles and online content within our corpus, and examine how it addresses dynamics related to ethnic identity. On this matter, Van Ginneken (1997) is quite prescriptive: "the production of meaning is intricately embedded in the activities of men and women in the institutions, or organization, and professions associated with their activities, and they produce and reproduce, create and recreate" (Van Ginneken 1997, p. 18).

To better understand the act of designation in a newspaper and in the online content of our corpus within the scope of this study, it is essential to consider the mediation of journalists or webnotes, who update designative referents used to define ethnic identities, as well as the (identitarian or non-identitarian) relationship that connects them to their source. Identifiable identities in publications (online or otherwise) are only through words used to designate them. In other words, they are the product of a journalistic naming protocol that has consequences in our perception (Calabrese 2008). Through the mediation of journalists and webnotes, they can control and neutralize the effects of these protocols (Boyomo Assala 2009; Tuchman 1972). This is why designations in a newspaper or in online publications sometimes bear the marks of a struggle for the definition of the meaning to be given to current events. The act of designation is thus intimately linked to the editorial policy of a newspaper and in the online discussion forums of Cameroon. It is the arena of social struggles (Siblot 1998). Its analysis reflects how the figure of the sender unfolds in the semiotic space of a given medium or platform, betraying intentions, goals, and the sought-after effect. This is made possible through how the contents of publications are handled in their enunciative structure. In our analysis, we specifically focus on event designators and actor designators within the studied printed and online publications.

## 2.1. Event Designators

Event designators come into play when a new event emerges on the public stage, with the media (or webnotes) appropriating it through a process of designation (Calabrese 2008). Through a series of operations, they seek to objectify it by defining its context and highlighting distinctive characteristics that facilitate its referencing (Ibidem).

Depending on the media's objectives or the webnote's intentions, the preferred characteristics in the designating term may include a date, a proper name, or even a strict event name (Calabrese 2009) containing semes of eventuality (conflict against Boko Haram). Based on this, we refer to event designators as discourse elements used to describe, and sometimes define, situational contexts through their strong referential and semantic load, sometimes linked to a historical-media event (Calabrese 2008, 2009). Their circulation through the media and continued presence in our daily conversations ensures their recognition throughout their duration in discourse (Calabrese 2008). They have the capacity to store the coordinates of the event to which they refer (Calabrese 2009). Shutz (1994) refers to this as a social stock of shared knowledge. Simply mentioning their names is enough to trigger the recall of the events they suggest. "The media thus give a name to the event, which is supposed to describe it (even in its future and transformations) based on the dominant values in society. Some have a high degree of formatting (elections, heatwaves), others imply a certain interpretation of the context (a crisis, a war, riots), and a small number involve the public positioning of the speaker regarding a conflict (a genocide)" (Calabrese 2009, p. 3).

Although the events suggested in our study by event designators did indeed have a physical reality, their verbalization is a socially constructed artifact. There are several types of event designators, including toponyms (which focus on the event's location as the main information coordinate), xenisms (aiding in the recall of toponymic coordinates through phonetic resonance and the exact naming of the event), and hemeronyms (event designators focusing on the date), etc.

In our research, we will delve further into hemeronyms. Morphosyntactically and semantically, the hemeronym is a date that designates an event (Calabrese 2008). Its

eventuality load relies on an evocative context, and its formulation fulfills a need for linguistic economy aimed at producing a specific effect on the representation of the event (Calabrese 2008). Pragmatically, the hemeronym is capable of designating events limited in time, with a strong political component that interrupts the usual course of society. It would thus have a very particular effect on the construction of the event, interrupting a natural temporality and establishing a transversal series of facts that, however, are not perceived homogeneously (Calabrese 2008, p. 6). In other words, it is a form of event designator whose explanatory and predictive character is operational in the study of events of a political nature (Calabrese 2008). In our view, this gives it sufficient relevance as an analytical instrument for examining the identity markers at play in the Cameroonian media space. The hemeronym that captures our attention in the analyzed articles is the date of April 6. It interests us here for the narrative it suggests in the national opinion, especially during the conflict against Boko Haram, specifically in a context where the attacks of this terrorist group are perceived by a certain political elite belonging to the Béti ethnic conglomerate as a rebellion in the North.

### 2.2. Actor Designators

Actor designators play a crucial role in framing events (Hilgartner and Bosk 1988, p. 58), as media outlets select information to disseminate and often assign designations to the actors they discuss. This task is carried out in reference to shared cognitive frameworks, professional protocols, and with the assistance of other social actors responsible for nomination and designation (e.g., the military, police) (Calabrese 2015). A few months after Cameroon officially entered conflict with the Boko Haram terrorist group, different media outlets used distinct designations for this organization. Some national media referred to it as a northern rebellion, a designation contested by others that earlier labeled it as Boko Haram, depicting it as a violent Islamist sect. In the former case, the designation attributes the aggression against Cameroon to a specific ethno-regional identity group, while the latter emphasizes an external aggression.

Hall (2007, p. 207) terms this situation the struggle for positionalities, highlighting the challenges related to designating social actors in newspapers within an environment of ethno-regional identity competition for power benefits.

## 3. Methodology

This research combines critical analysis of press discourse with sociological analysis of journalistic production. Our approach aims to contextualize the ideology behind the designations of press articles and online content under study, particularly by examining their relationship to the mechanisms they result from, such as (for printed press) the territorial anchoring of the newspaper, its editorial line, its language of production, and (for both printed press and online content) the identity of the person speaking, or the community involved in the event.

The main objective is to establish a dialectical relationship between the discourse of newspapers, online content, and the sociological foundation of their production. The analysis of press discourse is based on the examination of 497 articles published between 1 January and 30 June 2015. They are drawn from 7 Cameroonian newspapers selected based on several criteria: the importance given to the identity (ethnic) question and the social claims it has generated. Other criteria such as language of publication, territorial coverage, and the origin of capital (public, private) were also considered. Moreover, these newspapers hold significant symbolic weight in structuring the public space in Cameroon despite the crisis prevailing in this field of activity. These include the bilingual government daily *Cameroon Tribune* (CT) (with a circulation between 8000 and 12,000 copies), four private national dailies: two in French (*Le Messager* and *Le Jour*, each with a circulation between 3000 and 5000 copies), two in English (*The Post* and *The Median*, each with a circulation between 2500 and 5000 copies), and two regional newspapers (Table 1) (*L'Œil du Sahel* and *Ouest Échos*, each with a circulation between 2500 and 3000 copies).

**Table 1.** Variables for the analysis of print and online contents.

| Variable / Corpus | Territorial Anchoring of the Corpus | Editorial Line | Language of Publication | Individuals/Community Involved in the Event | Media Owner (Public/Private) |
|---|---|---|---|---|---|
| Print corpus | National, regional | ✓ | French, English | Individuals and community | Public and private |
| Online corpus | | ✓ | French, English | Individuals and community | Private |

Source: The author.

The online corpus includes 450 written publications from the three Facebook forums: *Le Cameroun c'est le Cameroun* (LCCLC), *Le Cameroun pour tous* (LCPT), and *Le Cameroun que je veux voir*. Our approach involved, initially, applying for membership in each of these online discussion groups. Once accepted as a member of these platforms, we conducted keyword searches in each group's search bar related to each of the news topics under study. Thus, in each of the Facebook groups, we initiated searches based on expressions such as war against Boko Haram, regional balance, and Anglophone problem. The goal was to see all the publications produced in each of these groups on these specific topics. In addition to the search bar, the Facebook social network allows the delimitation of a specific period in relation to the launched search. Our observation in this regard is that the three selected groups produced nothing on our analyzed news topics in 2015, 2016, and 2017. They were created at the earliest in 2018 for the oldest, otherwise in 2019 for the others. Although social networks are now a prominent source of information in Cameroon, this reality is very recent. With a population of nearly 30 million, Facebook was still one of the favorite social networks for Cameroonians in the first quarter of 2022. By February 8 of the same year, there were more than 5 million account holders. Facebook ranked third among the most visited platforms in the country, behind YouTube in second position and Google in first position.

For the group *Le Cameroun c'est le Cameroun* (LCCLC), which we studied, its oldest publications date back to 2018. We make this clarification because we identified four groups with the same name. After investigations with reliable sources, we learned that originally there was only one group with this name, created in 2015. The split into multiple groups would have occurred because the administrator of the original group had joined the MRC, a party whose candidate officially came second in the last presidential election of 2018. One of the groups resulting from this split is called *Le Cameroun c'est le Cameroun libre* (Cameroon is a free Cameroon).

Concerning the groups *Le Cameroun pour tous* (LCPT) and *Le Cameroun que je veux voir*, the oldest publications related to our research date back to 2019. On this basis, it was challenging for us to grasp the thoughts of internet users on our analytical themes within the study period from January to June 2015, which is our interval of investigation. We observed that several discussion groups of Cameroonians emerged between 2018 and 2019. Nevertheless, we chose to analyze this content as an extension of these news topics. The aim was to see if they are addressed through the same enunciative registers and if they carry the same imaginaries as in the print media, a few years earlier. The specific news topics that captured our attention are as follows: the fight against Boko Haram terrorism, the Anglophone crisis, and regional balance.

As for the sociology of journalistic production, it is based on 15 semi-directed interviews conducted with publishers, editorial managers, journalists who covered these events, officials from journalists' associations, and government officials dedicated to managing the media sector in Cameroon.

### 3.1. Data Processing and Analysis

To identify actor and event designations in our corpus, we plan to rely on what Gamson and Modigliani (1989) call packages. These are elementary cores of meaning, which can be a word, a date, or an expression with a designation value related to ethno-

regional identities, headlines, or simple statements. Our approach involves contextualizing the designations in our printed and online corpus in relation to the mechanisms they result from. This includes the territorial anchoring of the newspaper, editorial policy, language of publication, the identity of the media owner, and the person/community involved in the event (Table 1). The objective is to establish a dialectical relationship between the discourse of newspapers and the sociological underpinning of their production. Subsequently, we will indicate how and in what manner the designation mode deployed in our online and print corpora reflect changes in the dynamics of the Cameroonian media sphere (censorship, government repression, economic pressures, etc.) concerning the emergence of new voices in digital discursive spaces.

To conduct this work, we initially compiled the main designations of actors and events associated with the fight against Boko Haram terrorism in our print corpus, especially in the four identified event sequences. Next, we sought to understand how they were discursively framed in different newspapers and the sociopolitical stakes they convey. We then examined the online content corpus to determine if the structure identified in the print media for the fight against Boko Haram was replicated and to identify the event and actor designations whenever they were used in this context. The observed differences will help understand the changes in the dynamics of the Cameroonian media sphere (censorship, government repression, economic pressures, etc.) with the emergence of new voices in digital discursive spaces. The quality of the designation observed in the studied forums will determine the actual change or mere complementarity between print media and online content. Both spheres of content publication exhibit a significant asymmetry in the rigor of state control. Online content offers greater flexibility and a wider margin of maneuver, as authorities may not have the means to act online as they do in print media.

*3.2. Processing of Transcribed Interviews*

The objective behind processing the transcribed interviews is to account for the role of ethno-regional identity interferences in the editorial attitude or professional behavior of media actors (journalists and publishers). As mentioned earlier, information here proves to be less a product of journalistic work and more a result of interactions among journalists, webnotes, and other actors, particularly those from business and political spheres (Champagne 1984). These forces intertwine within the journalistic milieu, shaping its internal dynamics. Thus, the production of both print and online content is influenced by political realities. In this case, we consider ethno-regional identities within the context of the fight against terrorism.

Given the diverse profiles of our interviewees and the different positions from which each expressed themselves, maintaining a certain distance from the interview content became necessary. The goal was to conduct an analysis as detached as possible from the biases that characterize the statements of our interviewees. In such a context, our reflective approach and distancing in the analysis of interviews involved, each time, specifying the position of our interviewee, their background, and possible conflicts of interest that could influence their speech in the context of this research, essentially placing them within the broader context of the mechanisms they are part of. From there, we sought to better understand the relevance of their statements to produce our analysis in relation to the phenomenon under study.

## 4. Results

To conduct this work, we initially compiled the main designations of actors and events associated with the fight against Boko Haram terrorism in our press corpus and then online. Next, we attempted to understand their discursive framing where they appeared and the sociopolitical stakes they convey. The tables below (Tables 2–5) present the main designations of actors and events in our corpus related to the fight against Boko Haram terrorism, organized by newspaper.

**Table 2.** Designations of actors and events within the government daily in the context of the fight against Boko Haram terrorism.

| Newspaper \ Sequences | Population's War Effort | | Motions of Support for the Head of State | | Marches and Mobilization in Support against Boko Haram | | Theory of a Northern Conspiracy | |
|---|---|---|---|---|---|---|---|---|
| | Actor Designation | Event Designation | Actor Designation | Event Designation | Actor Designation | Event Designation | Actor Designation | Event Designation |
| *Cameroon Tribune* | Bamboutos Chiefs, The Elite of the North-West, The Haut-Nyong élites | War effort, popular support | The elites of Wouri Bwele canton, The driving forces of Menoua, The vibrant forces of Noun, etc. | | Southwest Youths, Inhabitants of the South, Bui Women, The elite of the South, The Noun March, etc. | Patriotic March | The Population of the Far North Region, the Eldest Daughter of the Renewal Movement | The Call of Lékié |

Source: The author.

**Table 3.** Designations of actors and events within the Anglophone press in the context of the fight against Boko Haram terrorism.

| Newspaper \ Sequences | Population's War Effort | | Motions of Support for the Head of State | | Marches and Mobilization in Support against Boko Haram | | Theory of a Northern Conspiracy | |
|---|---|---|---|---|---|---|---|---|
| | Actor Designation | Event Designation | Actor Designation | Event Designation | Actor Designation | Event Designation | Actor Designation | Event Designation |
| *The Median* | Southwest Chiefs, Northwest Fons | Solidarity against Boko Haham | Southwest Chiefs | | Persons from all, Cameroonians | Solidarity against Boko Haram | | |
| *The Post* | Southwest The Fako Chiefs | Solidarity against Boko Haham | | | Southwest | Patriotic march | | |

Source: The author.

**Table 4.** Designations of actors and events within the regional press in the context of the fight against Boko Haram terrorism.

| Newspaper \ Sequences | Population's War Effort | | Motions of Support for the Head of State | | Marches and Mobilization in Support against Boko Haram | | Theory of a Northern Conspiracy | |
|---|---|---|---|---|---|---|---|---|
| | Actor Designation | Event Designation | Actor Designation | Event Designation | Actor Designation | Event Designation | Actor Designation | Event Designation |
| *L'Œildu Sahel* | The elite of the Adamawa, the elite of the Grand North | War effort | | | The average Cameroonian, the minister, parliamentarians, ordinary citizens, United for Cameroon collective | Patriotic march, Kousseri march, Peul, Kirdi, Arab-choa | Nordiste | Northern conspiracy, Northern Rebellion, 6 April 1984 |
| *Ouest Échos* | The department of Koung-khi, High Plateaus, etc. | War effort | | | Le Haut-Nkam | | | |

Source: The author.

**Table 5.** Designations of actors and events within national private dailies in the context of the fight against Boko Haram terrorism.

| Sequences / Newspaper | Effort de Guerre des Populations | | Motions de Soutien au Chef de l'État | | Marches et Mobilisation de Soutien Contre Boko Haram | | Théorie d'un Complot Nordiste | |
|---|---|---|---|---|---|---|---|---|
| | Actor Designation | Event Designation | Actor Designation | Event Designation | Actor Designation | Event Designation | Actor Designation | Event Designation |
| le jour | The South region, the Nyong-et-Kelle department, elites of the Mfoundi, etc. | Effort of war, popular support, popular mobilization | | | The average Cameroonian, the minister, parliamentarians, actors from civil society, ordinary citizens, the collective United for Cameroon, etc. | Patriotic march | North, northerner | Conspiracy theory |
| Le Messager | The Bamboutos, The elites of the Sanaga Maritime, etc. | War effort, popular support, popular mobilization | The Bamboutos | | The Cameroonians, the minister, parliamentarians, Unis pour le Cameroun collective | Patriotic march | The minister, the president of the National Assembly | The call from Lékié, the specter of an April 6th, a show of strength from the northern front, problème nordiste |

Source: The author.

### 4.1. Designation of Actors and Events in the Print Media of Our Corpus

In analyzing these tables, we observe that the mode of designation of both actors and events in the fight against Boko Haram in Cameroon varies depending on the newspaper, its ownership (public, private), territorial coverage, editorial line, and language of publication (Table 1).

In privately-owned newspapers published in French with national territorial coverage (*Le Jour* and *Le Messager*), the mode of designation of actors and events relies primarily on toponymic referencing of actors, most of whom belong to the bourgeois ethno-political elite. For instance, in the sequence titled effort de guerre des populations (war effort of the populations), while the designation of actors almost exclusively refers to a small bourgeois ethno-political elite, the event is endowed with a popular aura. In *Le Jour* and *Le Messager*, synonyms such as soutien populaire (popular support) and mobilisation populaire (popular mobilization) are used, ultimately confirming the political crowning of a few territorially entrenched ethno-political leaders. Examples include Bamboutos Chiefs, l'élite du Nord-Ouest (the elite of the Northwest) (Table 5), les élites de la Sanaga maritime (the elites of the Sanaga maritime) (Table 5), and so forth. This type of designation associates a community or a personality with a specific territoriality, indicating their origin.

While this approach is somewhat measured or limited in privately-owned press (Table 1) due to the interests it carries, in the columns of the publicly-owned newspaper *Cameroon Tribune*, it is a nearly systematic approach. This newspaper was, in fact, most solicited by these ethno-political elites themselves to cover this news. The idea behind this strategy is to showcase the support of the diverse ethno-political high nobility in the fight against terrorism. In this event sequence, the designation of actors primarily associates the toponymic reference of the designated actor (department, region, canton, etc.) with their social rank/status (elites, traditional Chief, minister, etc.) (Table 2). This results in actor designations such as les forces vives du Noun (the living forces of Noun), les élites du canton Wouri Bwele (the elites of Wouri Bwele), and so on.

In the sequence titled théorie du complot nordiste (theory of the northern conspiracy), on the other hand, the designation of actors and events revolves almost exclusively around two communities: the Béti (from the Lékié department) and the northerners. In a statement called l'appel de la Lékié (the call of Lékié), which went viral and was published in the state-owned daily, the ethno-political elite of the Lékié department uses the "expression complice de Boko Haram dans les régions septentrionales"[1] (accomplice of Boko Haram

in the northern regions) to designate their colleagues from the northern regions of the same ruling political party. This situation recalls the painful past that links these two ethno-political communities, notably with the failed coup of 6 April 1984. *Le Messager* even speaks of the "spectre d'un 6 avril"[2] (specter of April 6) (Table 5) and of a problème nordiste (northern problem) (Table 5). The designation of actors in this sequence mainly relies on the toponymic references of the actors, using terms such as la Lékié, Nordiste, and régions septentrionales (northern regions) (Tables 2 and 5). It also relies on conflict markers with toponymic references, using expressions like "complices de Boko Haram dans les régions septentrionales",[3] (accomplices of Boko Haram in the northern regions), thus indicating a conflictual perception of this conflict with an identity anchoring.

In the publicly-owned newspaper, on the other hand, the designation of actors or events on this subject explicitly does not reference violence towards any specific ethno-political group nationally. The tone is not one of stigmatization. To refer to this territoriality, the newspaper speaks of la fille "ainée du Renouveau"[4] (the eldest daughter of the Re-newal), as if it were in a courting situation. For many political observers, the leaders of Boko Haram reportedly find it relatively easy to recruit from the youth in the Extreme North region simply because it is perceived as neglected by the public authorities: under-education, economic poverty, insufficient healthcare and road infrastructure, among other grievances. When the state-owned newspaper refers to la fille ainée du Renouveau in this manner, it adopts a seductive and counter-communication approach. The aim is to convey that this region is at the heart of the presidential attention and has been since the arrival of Paul Biya in power. These explanations are explicitly clarified by the newspaper in the following excerpt:

> "The political elites, regularly echoed by the media, often like to repeat, at their leisure, that the Far North, emerging in August 1983, in the aftermath of the breakup of the vast province of the North, which covered at the time the three current northern regions (Adamaoua, North, and Far North), is the eldest daughter of the Renewal. This expression is not an empty slogan devoid of any foundation. By traversing the deep Cameroon, the Far North can boast of being among the privileged regions of the country in terms of roads: all departmental capitals are connected to the regional capital by paved roads. . ."[5]

However, the newspaper fails to mention that the fragmentation of the former large Northern region into three, including the one affected by Boko Haram, was more of a political maneuver. It aimed to weaken the political base of the first president, where he still had significant support.

In the French-language private newspapers, the event designation in this sequence known as the theory of the northern plot also relies on references that suggest conflict.

This is evident with expressions like the rebellion of the North, the northern plot, the insurrection of the Grand North, and northern problem. This is a lexical field of aggression associated with an ethno-political community. This event designation also takes the form of a date that acts as a synonym for the same aggressive reference, such as the specter of an April 6, the date of the failed coup. This is mainly developed in the regional newspaper *L'ŒIL du Sahel* but also in *Le Messager*. However, this lexical field of aggression is completely changed in the other subsequent event sequences, namely the war effort of the populations, motion of support for the Head of State, and march and mobilization in support of the army. In terms of the number of publications for each journal on these topics, the first two did not interest these editorial teams much. As we have pointed out, the ethno-political elites of each locality had sought the public capital newspaper more for this coverage. Nevertheless, for the few designations of actors that come back to it, we observe fewer conflictual references as in the theory of the northern plot, even if the designation of actors remains mainly characterized by toponymic references.

Within the private English-language newspapers (Table 1) in our corpus, this event sequence has not been the subject of many publications. It is as if it were a Franco-French question. However, this is indeed a question of power struggles at the top of the state.

The only articles published by the English-language press (Table 1) in the corpus on this subject exclusively focus on the statements of the Chadian Minister of Communication, who reveals that the majority of weapons found in the hands of Boko Haram militants are of French origin, without pointing an accusing finger at a specific Cameroonian ethno-political group as an accomplice. Likewise, for the three other event sequences of the conflict against Boko Haram (war effort of the populations, motion of support for the Head of State, march, and mobilization in support of the army), these newspapers almost exclusively focused on what happened in the Anglophone regions and by actors originating from these territories. Yet, we are dealing with newspapers that are not regional but national (Table 1). The main designations of actors found here in the context of the war effort of the populations are Southwest Chiefs, Northwest Fons (Table 3). These are elitist references that are mainly found in reports and summaries. However, this event, in the columns of Cameroun Tribune, has interested several other ethno-political and linguistic leaders in territorialized regions. We observe this mode of coverage also in support marches and motions addressed to the Head of State in these newspapers—a coverage mode that we qualify as Anglo-centric for newspapers that claim a national territory.

In the private press with regional territoriality (Table 1) in our corpus (*L'Œil du Sahel, Ouest Échos*), the relationship to news about Boko Haram terrorism or the number of publications seems to depend greatly on the territory of the newspaper and the ethno-political community involved in the event. We see this, for example, in the designation choices in the sequence reserved for the theory of the northern plot. In *Ouest Echos*, for example, there is no mention of a northern plot, rebellion of the North, let alone April 6 or northern problem. On the other hand, these designations return in *L'Œil du Sahel* with the perspective of deconstructing them, with articles of analysis and extensive interviews often granted to ethno-political leaders in this region. After the publication of the communique called the call of the Lékié in the government daily, the newspaper Nordistes will specifically focus on this chosen piece of this document.

> "Not against all pernicious manoeuvres on the part of these accomplices of Boko Haram; manoeuvres of political blackmail comparable to an attempt to take hostage or destabilise the Institutions of the Republic or to a political conspiracy, inspired by various ends, in particular political, personal or regionalist ambitions; against the personal and unspeakable behaviour of these objective allies of Boko Haram, in particular their doublespeak and other similar failings, with regard to the fundamental and permanent obligation of loyalty, which is incumbent on all citizens, at all levels, towards the President of the Republic."[6]

In its analysis of this statement, the newspaper first addresses the disassociation of other power elites belonging to the Béti ethnic conglomerate. Specifically, the Minister of Higher Education and the then Secretary-General of the Presidency, both influential figures in the Bulu tribe, the same as the Head of State. In the local political context, this reflects the notion that within the broader Béti community, these accusations do not enjoy unanimous support, especially among the Bulu elite. The article notes, "this literature had sparked multiple reactions, all condemning such deviations in the Republic. Jacques Fame Ndongo, Belinga Eboutou, and many other personalities distanced themselves from this stigmatizing position."[7]

It is not coincidental that the newspaper focuses precisely on these two political heavyweights originating from the South region. The first has held the position of Minister of Higher Education for over 20 years and is also the regional leader of the ruling RDPC party in the South. The second, who has since passed away, was, at the time of the events, the Director of the Civil Cabinet of the Presidency. This underscores the significance of the ethnic variable in Cameroonian politics and its influence on the editorial choices in covering certain events. This could be seen as an ethno-political bias that is not lost on the journalists at this publication, who are aware of the stakes surrounding this issue.

In *Ouest Échos*, there seems to be a general lack of interest in this news. The ethno-political groups primarily in conflict in the coverage of this news are not within its coverage territory. However, this is a national issue, especially concerning the fight against terrorism.

*4.2. Designation Issues Vary from One Newsroom to Another*

- **Journalists and political players use different methods of appointment.**

We saw this in *Cameroon Tribune*, the state-owned daily. This state-owned daily refuses to use the word 'Nordiste' in the texts written by its journalists. Nor is there any mention of a conspiracy. The newspaper prefers to use official toponymic references or even political flattery to refer to the people of this part of the country stricken by Boko Haram in this sequence of events. It refers to the people of the Far North region, and even to the eldest daughter of the Renewal[8] (two issues), as can be seen in this extract:

> Political elites, regularly echoed by the media, like to repeat at will that the Extreme North, which emerged from the baptismal font in August 1983, following the break-up of the vast Northern Province, which at the time covered the three present-day northern regions (Adamaoua, North and Extreme North), is the eldest daughter of the Renewal. This expression is not a hollow slogan devoid of any foundation.[9]

The excerpt is from an article in which the newspaper defends the government against criticism claiming that terrorism is thriving in the region due to rampant poverty and neglect of the population. In its edition of Tuesday, 2 September 2014, Cameroun Tribune published a list of statements supporting the Head of State. The most resonant was the one called Lékié Appeal (Table 2). It is signed by high dignitaries of the government, characterized by belonging to the Eton and Manguissa ethnic groups (from the Béti-Bulu ethnic conglomerate). They accuse political leaders from ethno-political and linguistic groups originating from the Grand North, largely Islamized, of being *"accomplices of Boko Haram, mainly in the northern regions of Cameroon, and of their sly strategies or attempts to incite the partition of the national territory, in light of the regrettable developments recorded in other countries or regions of the African continent."*[10] A stigmatizing designation mode that contrasts with that of the journalists in the editorial office. There is no room for censorship in this type of content in the government newspaper, even when the statements undermine social cohesion, especially when made by top government officials. While the state-owned newspaper, through the publications produced by its journalists, does not directly accuse northern elites or use stigmatizing designations, it leaves this latitude to ethno-political leaders of other identity groups. The newspaper has not faced any sanctions from the National Communication Council, the regulatory institution for the media. Similarly, no editorial manager has been sanctioned for this publication resembling hate speech. This indicates that Cameroun Tribune's publications are the product of a media environment dominated by the political authorities. To illustrate this political influence in the newspaper, the Editor-in-Chief revealed, *"Cameroon Tribune is a newspaper where everyone thinks they have a right to oversight. As soon as you have a position, if you are the President of the National Assembly, for example, you believe that you should make sure there is also someone from the Far North at Cameroon Tribune".*[11]

This reference is not at all insignificant, considering that the President of the National Assembly is indeed from the Far North region. This situation suggests various layers/surfaces of embedding and power relations that intersect with each other in this editorial office, particularly ethnic power relations often associated with language, region, tribe, etc.

- **6 April: Between commitment and disinterest, depending on the community involved in the event**

Within our corpus, the date of 6 April 1984 (Table 5), in reference to the failed putsch, is actively evoked by three newspapers: *Le jour, Le Messager*, and *L'Œil du Sahel*. In the columns of *Le jour*, the date is cited eight times in seven issues of our corpus.[12]

In *Le Messager*, this date is mentioned six times in two issues.[13]

The newspaper devotes an entire dossier to it, notably in its issue number 4298 on Tuesday, 7 April 2015, on page 4. In *L'Œil du Sahel*, this date is cited 19 times (Table 4) in our corpus,[14] including 12 times in issue number 697 on Thursday, 23 April 2015. In the edition of Thursday, 2 April 2015, on page 9, we learn on this subject that

> ultimately, essential questions that only historians will elucidate one day remain today. Indeed, many people continue to wonder, for example, if the coup was not desired or provoked, to find a credible and solid pretext to clean up the remnants of what would later be called the old regime. If that were the case, it cannot be said that this strategy was not successful. Whether it was a trap or a solitary adventure of a handful of misguided military personnel, it is undeniable that the northerners have never recovered from this historical accident. Since 6 April 1984, they continue, in one way or another, to pay for the mistakes of a few of their own.

In other words, for this newspaper, it is a story whose official intrigue's initiative remains in the hands of the victors. In English-language newspapers (Table 1), despite their national focus, there is no mention of these events. This gives the impression that, from the perspective of Anglophones, it is a Franco-Francophone affair. In the columns of *Cameroon Tribune* and *Ouest Échos*, very few references to these events are found. In the government-owned daily, we only found one mention of this date. It is a column by a communicator from the ruling party. He expressed outrage at the media treatment by French newspapers towards the Head of State. He declared, in essence, to denounce it:

> (. . .) how can anyone be unaware, with reference to what is stated in the article in Le Monde Afrique, that the BIR did not exist at the time of the failed coup d'état on 6 April 1984, but only since 2001?[15]

Not surprisingly, the state-owned newspaper never revisits these events, as its editorial line aligns with the official positions of the government. In *Ouest Échos*, this date also appears once incidentally, in an open letter critical of the Head of State. The letter is written by a Cameroonian expatriate from Noun, from the Bamoun ethnic group in the West Region of Cameroon, as specified in the newspaper. One can read, in essence:

> (. . .) Colonel NCHANKOU (President of the military tribunal during the events of 6 April 1984) and today Mr. YAP ABDOU, president of the special criminal tribunal (as part of Operation Sparrow). These few examples show the effectiveness of the Bamum officials in carrying out the tasks entrusted to them by the Head of State. It is therefore curious that all the Bamum are now being punished because they are assimilated to UDC militants.[16]

Considering the above, there is selective attention to events suggested by the April 6 date, depending on the newspapers. This selection evidently varies based on the language of the newspaper, its territoriality, and its editorial line. Only privately-owned French-language newspapers revisit this date. Among them, there is a publication specializing in the northern regions, and two nationally renowned newspapers known for their criticism of the government. This reveals a fragmentation of the media public space. This fragmentation is based on politico-ideological, as well as ethno-regional and linguistic factors. The Publisher of *L'Œil du Sahel,* who is also a native of the territory covered by his newspaper, does not deny his proximity to political leaders in the Grand North. A similar attitude was found in his colleague from the West Region, also a native of the territory covered by his newspaper. Hence, this date may seem less relevant to this editorial team because the events it conveys relate to another region and, consequently, to another ethno-political group. In the interview that the Publisher of this editorial team granted us, he emphasizes the precedence of his role as a politician over that of a journalist. He states, in essence:

> I'm a politician even before I'm a journalist, because on campus, at the time of the democratic opening, I was the coordinator of the students of the Union des Populations du Cameroun (UPC), which is the first political party in Cameroon,

the nationalist party. So I was the coordinator of the UPC students at that time and I was the first president of the congress of the youth branch, the young people of the JDC, which is called the JDC. Jeunesse Démocratique du Cameroun, which is the youth branch of the UPC and at that time I was the first president who organised the congress after coming out of hiding. As you know, the UPC was banned in 1955, temporarily rehabilitated in 1962 with Mayi Matip before being reabsorbed into the Cameroon National Union, the single party that was created in September 1966. So when democracy was opened up, we came out of hiding and I was the president who led the young people of the UPC to their first congress in 1992.

In other words, although *Ouest Échos'* editorial line would, in principle, justify the lack of attention to the April 6th date in this editorial team, this choice is also based on political considerations.

### 4.3. Designation of Actors and Events in Our Online Corpus

The online corpus is highly dynamic and undergoes continuous changes. On the Facebook social network, it is even more dynamic, as users can delete their posts. Additionally, the social network provides a wide variety of content appreciation modalities, including comments, likes, dislikes, and more. Regardless of the publication year of a content piece, modifications can occur. Therefore, it is challenging to have stabilized content on which we can consider a representative sample. Similarly, for each of the event sequences studied in the fight against Boko Haram, we realized that users' posts and discussions do not necessarily align with the event sequencing in the print media. While the same thematic event (the fight against Boko Haram) seems to be shared in both print and online media, in the online sphere, users appear to engage more freely in choosing publication angles and discussions. Thus, the fight against Boko Haram is not necessarily perceived here in the event sequences of war effort of the populations, northern conspiracy theory, march and mobilization in support of the army, or even motions in support of the Head of State, etc. The dynamics of users' posts are often influenced by current events and sometimes vice versa. Having content on the Boko Haram conflict was sporadic/episodic based on new information on the subject. Indeed, while most of the conflict favored the Cameroonian army, there has not yet been an official declaration of the end of the conflict against Boko Haram by the Cameroonian Head of State. Between 2015 and today, Boko Haram has shifted from coordinated and organized frontal attacks to high-profile suicide bombings. In 2015, at the height of the conflict, the interest in social media as information hubs in Cameroon was not as prevalent as it is today. This phenomenon significantly gained traction during the 2018 presidential election.

4.3.1. Online Designation in the Fight against Boko Haram: A Tool for Inter-Ethnic Hate Speech

By hate speech, we mean a malicious discourse motivated by prejudices, targeting an individual or a group based on their inherent real or perceived characteristics. It expresses discriminatory, intimidating, disapproving, antagonistic, and/or harmful attitudes towards these characteristics, including gender, race, religion, ethnic background, color, national origin, disability, or sexual orientation. The purpose of hate speech is to injure, dehumanize, harass, intimidate, weaken, degrade, and victimize targeted groups, fostering insensitivity and brutality towards them (Cohen-Almagor 2011, pp. 1–2).

From this definition, three dimensions of this type of discursive manifestation emerge, namely the affective dimension, primarily an emotional and sentimental expression. It then acts as a stimulant for violence towards others, and finally, a powerful consolidator of stigmas (Monnier et al. 2019).

In Cameroon, the changes in the dynamics of the Cameroonian media sphere (censorship, government repression, etc.) with the emergence of new voices in digital discursive spaces, manifested by the migration of political discourse from print media (subject to

state regulation) to online forums (outside the control of the state), have led to the rise of inter-ethnic hate speech around power issues.

By way of comparison, the conflict against Boko Haram in the print media has reported on several categories of actors, including the national army, Boko Haram, ethno-political leaders from each territory, and the Head of State. In print media, this political conflict primarily opposes a part of the Elite from the Beti ethnic conglomerate, especially those from the Lékié department, to the ethno-regional Elite of the Grand North of Cameroon. The former accuses the latter of being behind Boko Haram with the support of a major power, with the idea of seeking revenge for the failed coup d'état on 6 April 1984.

Conversely, in the online content of the studied forums, discussions about the conflict against Boko Haram are more focused on criticizing the Cameroonian army, referred to as a 'Bulu militia,' an ethno-political critique of the incumbent regime associated with the Beti ethnic group (to which the president belongs). Boko Haram attacks are more of a pretext for this criticism. All of this is articulated in an electoral context that witnessed the emergence of the Beti/Bamiléké divide. The electoral aspect of these online ethno-political clashes between Beti and Bamiléké, under the false pretext of the conflict against Boko Haram, is clearly illustrated in the publication below, taken from the '*Le Cameroun pour tous* (LCPT)' forum. The oldest publications from this group date back to 2018, a year marked by a tumultuous presidential election that saw the resurgence of the divide between these two communities. Both communities resort to derogatory designations, symbolizing an escalation of ethnic hatred in online politico-media discourse. The forum's administrator (referred to as Éric le Retour) published a clarification note on 2 November 2018, to specify the objectives of the initiative, as captured below (Figure 1).

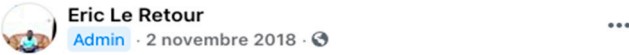

**Figure 1.** Screenshot of the 2 November 2018 post by Éric le Retour in the Facebook group *Le Cameroun pour tous* (LCPT), 20 January 2023 at 2:35 a.m. Translation into English: "Settlement of accounts! We are fighting against the RDPC, not the Beti people. Wilfried Ekanga cannot fight against the culture to which he belongs; Fabrice Noah Noah cannot struggle against his own tribe, and Sorelle Onguene cannot be against her own brothers. Our Beti brothers who, naively or intentionally, play on the identity card to obscure the truth of the election results, will have only themselves to blame. We are fighting against the RDPC, which has done next to nothing for Cameroonians for 36 years. Too bad for the sorcerers. In 2011, when they sensed that Marafa was eyeing power, they opened the door to Boko Haram and let the people know that the northerners wanted to destroy Cameroon. Today it's the Bamileke. We'll see!".

In this publication, the author draws a connection between political parties and ethnic identity. These links had already been highlighted by Menthong (1998) about the 1992 presidential election. But he goes on to try to dissociate ethnicity and political party when he states in substance that "We are fighting the CPDM, not the Béti people". According to this publication, the conflict against Boko Haram terrorism is defined as a political weapon deployed in the context of the 2011 presidential election by Béti elites to neutralize those of the Far North. A scenario that seems to be repeating itself in the 2018 presidential election,

just a few days before this publication. An election in which outgoing President Paul Biya, of Béti origin, faces a candidate of Bamiléké origin as his main challenger. This explains his last sentence: "Today it's the Bamiléké. We'll see!", effectively associating political party and ethnic identity. The designation of actors in his publication considers ethnonyms (Bamiléké, Béti). It also uses ethno-political toponyms (les Nordistes), which it articulates in a conflict dynamic. Yet this discussion group has set for itself the rules of friendliness, courtesy, refusal to incite hatred, and so on, as mentioned in their rules and regulations below, taken by screen capture (Figure 2).

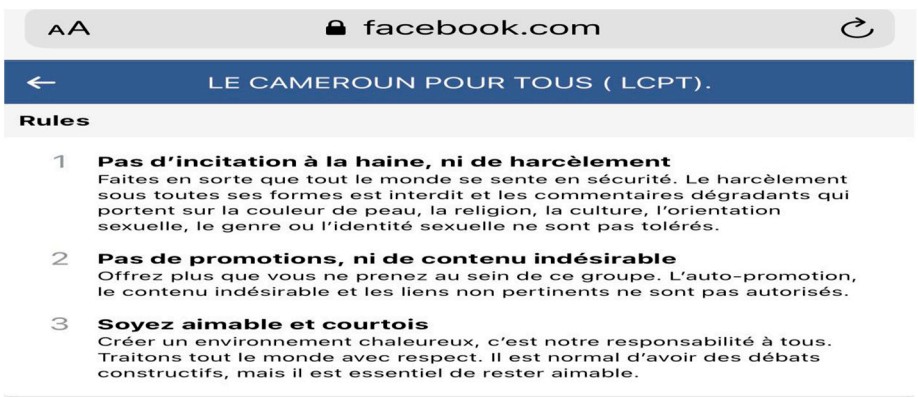

**Figure 2.** Screenshot of the rules of the Facebook group *Le Cameroun pour tous* (LCPT), 20 January 2023, 2:37 am. Translation into English: "No incitement to hate, no harassment, no promotions or unwanted content. Be kind and courteous."

The administrator himself is breaking the rules he has set himself. In terms of our sequencing of events, we can possibly classify this publication as part of the fight against Boko Haram. The author alludes to the northern conspiracy theory, but this time it is the Béti versus the Nordiste conspiracy for electoral purposes. Indeed, the brief reminder of the northern conspiracy theory is used in this publication to indicate that the regime in power (which it equates with the Béti in this case) always uses identity-based divisions to win elections. This is a technique whereby the regime in power victimizes itself on the pretext of being attacked by a community that is supposed to be a rival in terms of its political and identity background. In this way, the regime would use its media to justify its victimization of a community that was either rebellious or uncontrollable. This would lead to a war-mongering vocabulary in the above publication.

This is the case with words and expressions such as nous combattons; Voulait détruire le Cameroun; Lutter; S'acharner; (We fight; Wanted to destroy Cameroon; Struggle; Persist.), and so on. Words and expressions are articulated with ethnonyms such as Béti, Bamiléké, political party names and ethno-political toponyms, i.e., designating actors with territorial referents (northerners). This publication also met with a great deal of sympathy from the other members of the discussion group regarding the evaluation methods shown below in the screenshot (Figure 3).

As we can see, no one expressed disagreement, symbolized by the upside-down blue thumb. On the other hand, the right-side-up blue thumb was used 61 times, the remainder of the appreciations leading to the number 73 being distributed between the other forms of appreciation not considered for our analysis. In reading this publication, we go back to Freitag (Dagenals 2003) to say that uses of the Internet contribute to the reinforcement and (re)production of a hierarchy of power. In the case in point, it redraws the identity cleavages that we already knew from the printed press and the ethnic groupings that Paul Biya (1987, 2018) was already observing in the streets of Cameroonian towns more than thirty years ago before digital social networks. The reality of Cameroonian political events in online discussion forums would therefore be no more than a cybernetic extension of material reality.

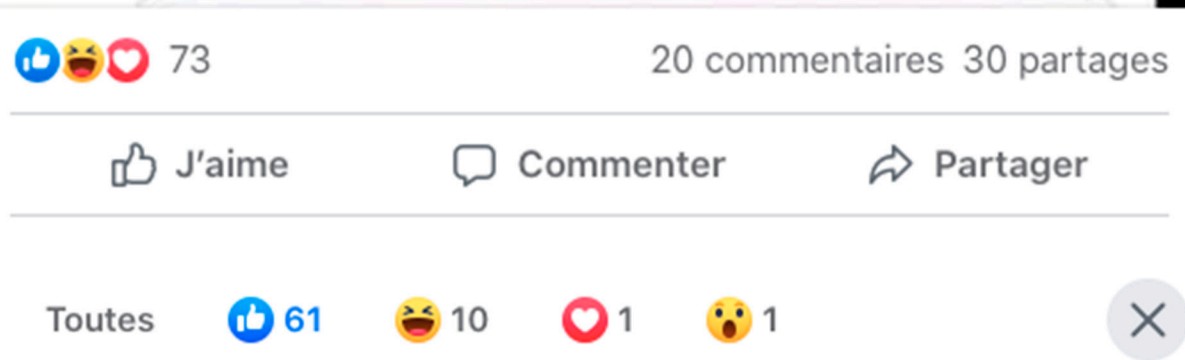

**Figure 3.** Screenshot of appreciations of the 2 November 2018 publication by Éric le Retour in the Facebook group *Le Cameroun pour tous* (LCPT). 20 January 2023 at 2:39 am.

This clearly shows that journalists have lost their monopoly on defining the meaning of current events to Internet users. What is more, no reference is made to the ethno-political elites of Cameroon's Far North, as in the print media. Allusions to this conflict sooner give way to ethno-political hate speech between Béti and Bamiléké, all against an electoral backdrop. Something that would never have happened in the print media at this level of discursive violence. Cameroon's media regulatory bodies crack down on incitement to hatred. Online, however, it has free rein.

4.3.2. Designation of Actors and Events against the Backdrop of the Béti/Bamiléké Power Struggle

As a result of socio-political cleavages with strong identity roots, the way in which the conflict against Boko Haram is viewed in online content follows the contours of political adversity. This can be seen, for example, in this publication from 1 July 2019 by Mimi Ekosso. In it, she likens the government's current attitude to the conflicts in the country to that of the Béti community. The national army is described as a Bulu militia. In other words, made up exclusively of members of the tribe of the President of the Republic. This is notoriously untrue. For this Internet user, the conflicts Cameroon is experiencing (including the conflict against Boko Haram) are the work of this ethnic militia. In other words, whether it is the Anglophone crisis, the conflict against Boko Haram, or the insecurity in East Cameroon, for this Internet user it is simply ethnic warfare. Any possible new episode or even allusion to the conflict against Boko Haram is used for ideological purposes with an ethno-political aim. Below is a screenshot of his publication (Figure 4).

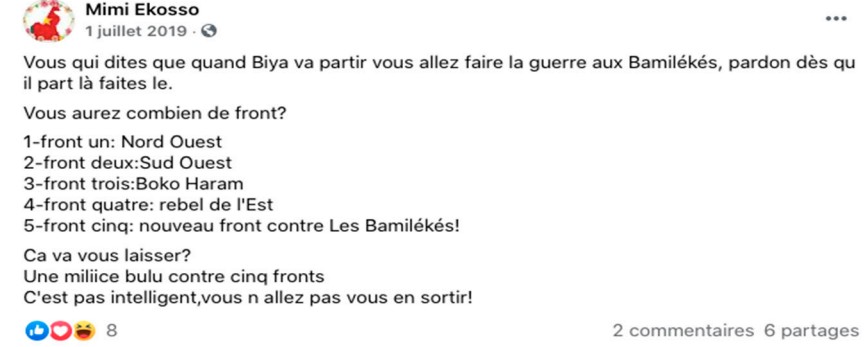

**Figure 4.** Screenshot of Mimi Ekosso's 1 July 2019, publication in the Facebook group *Le Cameroun c'est le Cameroun* (LCCLC). 20 January 2023 at 4:55 am. Translation into English: "Those of you who say that when Biya leaves, you will wage war against the Bamileke, please, as soon as he leaves, go ahead. You will have how many fronts: 1- Front one: Northwest 2- Front two: Southwest 3- Front

three: Boko Haram 4- Front four: Rebels from the East 5- Front five: New front against the Bamileke! Do you think you can handle that? A Bulu militia against five fronts. It's not smart; you won't get out of it!" Mimi Ekosso, Facebook, 1st July 2019.

This kind of speech would certainly never have been produced in the printed press. Cameroon's media regulatory bodies would certainly have cracked down on it for inciting hatred. But online he has free rein. For this Internet user, the conflict against Boko Haram is just one of the battles for a change of political regime in Cameroon. A change which, according to her, would be envisaged on an ethnic and military basis. The army itself is a Bulu militia, notably the ethnic origin of the Head of State. This gloomy publication of the future has not aroused much sympathy in view of the assessment methods below (Figure 5).

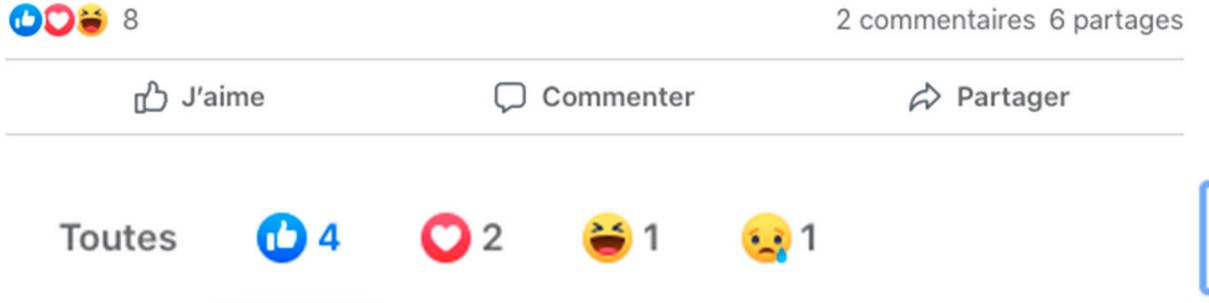

**Figure 5.** Screenshot of the terms of appreciation of the publication of 1 July 2019 by Mimi Ekosso in the Facebook group *Le Cameroun c'est le Cameroun* (LCCLC). On 20 January 2023 at 4:57 am.

Just four likes and two comments. On the other hand, six Internet users relayed it to other places of visibility. This publication clearly shows the level of radicalism that certain Cameroonian Internet users have reached in the cyber political game. In the controlled environment of the print media, we would certainly never have seen this kind of publication. Ethnonyms are articulated with a warlike vocabulary (Bulu militias, the war against the Bamiléké). This is a sign of a strong gradation of ethnic hatred, in this case exacerbated between the Béti and the Bamiléké.

Yet the directors of this group have set themselves a single rule—that of friendliness and courtesy. However, they left this publication and three years later it is still there. Below is a screenshot of the rules for this group (Figure 6).

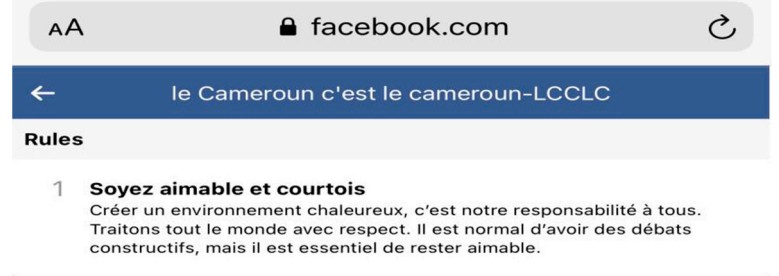

**Figure 6.** Screenshot of the rules of the Facebook group *Le Cameroun c'est le Cameroun* (LCCLC). On 20 January 2021 at 4:59 am. Translation into English: "Be kind and courteous. Creating a warm environment is our responsibility to everyone. Let's treat everyone with respect. It's normal to have constructive debates, but it's essential to remain kind."

## 5. Conclusions

In summary, the advent of social media in the dynamics of the Cameroonian media landscape has led to significant changes in journalism practices. It has spurred the emergence of new digital actors who now share the power to interpret news with journalists.

This study observed this phenomenon through the analysis of event and actor designations in both print and online publications in our corpus, within the context of the fight against terrorism in Cameroon.

The findings indicate that the rise of social media as new information platforms in Cameroon has transformed the designation of events and actors in the fight against Boko Haram terrorism, intertwining it with the ethnic identity issues of the sociopolitical environment. While this reality was already present in print media, its migration to online content has metastasized into hate speech. This phenomenon, which was once controllable through the actions of state regulatory bodies, now reflects a transition from a press discourse environment regulated by state institutions to a realm where the state has limited control means. Consequently, full censorship and other forms of government repression are challenging to enforce.

Describing the relationship between the media and the state under the old regime (1960–1982) in Cameroon, Thomas Atenga (2004) speaks of an environment where journalists had to primarily be patriots or even development agents who unquestionably adhered to the President's discourse. Journalists in Cameroon during this period were "conveyors of propaganda, sedative, calming, and soporific information in the name of the imperatives of nation-building, maintenance of public order, and the preservation of good morals. They were dedicated proselytizers, brokers of the 'father of the nation's' speech and the activities of his administration" (Atenga 2004, p. 38). In other words, journalists' work essentially conformed to a "grammar of flattery" (Atenga 2009, p. 72), where connivance prevailed, blending marketing, propaganda, and political advertising.

With the arrival of Paul Biya's new regime in 1982 and the liberalization of the press and associative movement in 1990, the Cameroonian state must now learn to deal with a press that can criticize and express opinions on certain decisions. Since the liberalization of the communication sector in the 1990s, establishing a newspaper in Cameroon requires very few administrative procedures.

This did not happen spontaneously. It first involved a nearly five-year struggle to transition from the authorization regime to the declaration regime, which was formalized in January 1996 through the amendment of the 1990 law endorsing freedom of communication. Until then, newspapers still faced the issue of prior censorship, a problem that was also abolished with the amendment of the same law. In this context, the state and the media maintain conflictual, even macabre, relations (Atenga 2004), within a sociopolitical environment described as authoritarian. Given the state apparatus's limited tolerance for criticism, a new era has emerged, witnessing an intensification of judicial repression against the media and journalists. Over 300 press trials were recorded in the early 1990s–2000s (Atenga 2004, p. 5). "The cumulative infringements on freedom of expression (prior censorship, seizures, suspensions, bans, raids on editorial offices, physical assaults, imprisonment of journalists, fiscal harassment, and various forms of intimidation) have far exceeded 2000" (Atenga 2004, p. 5). "The announcement of a presidential visit abroad or the Chief's vacation destination can lead to a trial for endangering state security, while in democracies, the president's agenda and movements, as well as those of ministers and other personalities, are made public days in advance... A journalist can be condemned by a court for the substitution or possession of administrative documents because he dared to present the judge with evidence of his allegations. The penalty is even more severe when he does not have them. In any case, he goes to trial already condemned" (Atenga 2004, p. 10). This helps in understanding the limited maneuvering room in the use of event and actor designations.

However, with the advent of Web 3.0 in journalism, a new type of information relationship is now dominant through digital social networks (DSN). This has resulted in the infiltration of hate speech into the political-identity game in Cameroon. Our research results have sufficiently highlighted this through the uses of actor and event designations by web users. These designations are articulated with political-identity ethnic issues, demonstrating that media designation in Cameroon is a site of power relations.

The theoretical foundations of the relationship between identity and media are rooted in the writings of Hall (2007, pp. 203–14), particularly in "New Ethnicity". According to Hall, the relationship between identity and media's discursive strategy is based on an ideological struggle, raising the question of "representation relationship" (Hall 2007, p. 204). Based on these approaches, the media shape our understanding of ethnicity, suggesting how it should be perceived, understood, and the meanings it conveys. This practice relies on significant effort in media representation. Hall emphasizes that "representation is only possible because enunciation always occurs in codes that have a history, occupying a position in the discursive forms of a specific time and space" (Hall 2007, p. 209). In other words, "we all speak from a place, a particular experience, a specific history, a particular culture. [...] And in this sense, we are all ethnically located, and our ethnic identities are crucial to the subjective sense of who we are" (Hall 2007, p. 210).

In the Cameroonian context, this perspective views ethno-regional identities as a source of power in the realm of political competition. In other words, belonging to a specific ethno-regional group would be a political asset. This identity-centric vision of political power is responsible for the transposition of the reading of ethnic identity into the Cameroonian media field. Due to the stakes related to political competition, Cameroonian newspapers are labeled based on ethno-regional identities. Therefore, "the process of determining events and situations does not solely rely on the media and involves a plurality of actors" (Arquembourg 2005, p. 29).

This approach refers to media designations as a mediation site among various actors, stakeholders in a social reality (Garric 2009). In Ricœur's terminology (Ricœur 1983), these are precisely narrative operations in the mimetic process, allowing an understanding of how informative narratives achieve their objectives (Arquembourg 2005).

Therefore, we conceptualize designations as sites of incorporation (Granovetter 1985; D'Hont and Gérard 2015; Le Velly 2002; Laville 2008; etc.) of power relations in society. The representational approach of the media often results from the intertwining or entanglement of various power relations operating in society (Bobo 1995; Collins [1991] 1990, pp. 68–69; Hooks 1992, 1996; Hall 2007). These can be identity-based, religious, cultural, racial, and often economic. Therefore, Le Velly (2002) apprehends incorporation as a sociology of market exchanges at the heart of social construction processes. Applied to the media domain, the logic of incorporation (Granovetter 1985; D'Hont and Gérard 2015; Le Velly 2002; Laville 2008; etc.) accounts for the fact that the interests of media actors and those of economic and political environments form a system. This would result in event coverage based on the agreement model (Boltanski and Thévenot 1991; Legavre 2011).

In the Cameroonian sociopolitical context influenced by the philosophy of communitarian liberalism (2018, 1987), this situation reveals various layers/surfaces of incorporation and power relations that intersect. Particularly, ethnic power relations, often associated with language, region, tribe, etc., further complicate things from an analytical perspective. It becomes challenging to ultimately identify what guides the editorial orientation of newspapers, including online content. Our perspective also considers designations as editorial processes crystallizing current events within frames of ethnic and territorial identity in a sociopolitical environment governed by competition on power issues. In other words, concerning the news studied in this research, designations of actors and events are primarily means of ethnic identification and territorialization in newspapers and online forums. To put it differently, the observed media designations in our corpus are also sites of ethno-political territorialization of the Cameroonian national space. Indeed, the territorialization of the Cameroonian national space is strongly oriented towards the identity diversity (ethnic and linguistic) characterizing the country.

This initiative is part of the state's desire for "territorial differentiation" (Pailliart 2013, p. 278). It is a long-term process, initiated by both the state and the media, aiming not only to designate the multiplication of territorial levels marking the country's map but also, more fundamentally, the movement of differentiation in the social world (Pailliart 2013). In Cameroon, the media heavily relies on the state's policy of territorialization. There are

national, regional, departmental, etc., media entities that contribute to mastering territorial differentiation and, by extension, social differentiation. It is in this aspect that we consider the territoriality of the newspaper in the analysis of the designation we conducted. That is, as a movement of differentiation of ethno-regional identity (Abondo Ndo 2022), an approach now strongly contested with the advent of social networks, whose territoriality is global. Thus, they challenge traditional obstacles such as censorship, government repression, and economic pressures, etc.

The way in which Social Media Networks (SMN) permeate the news studied in this research ultimately highlights the issue of media regulation in a multi-ethnic context in Africa. Ethnicity is a crucial variable around which power struggles are organized. Therefore, how should media regulation be reorganized in Cameroon in view of this challenge posed by SMN? Indeed, in addition to SMN, Cameroon is entrenched in a context of the end of a reign of over 40 years at the helm of the state, and where the press unfolds a "tribal war through intermediary newspapers [where] particularisms are thus magnified and defended. These are deviations that are justified, but when taken too far, can prove very dangerous. No longer are ideas or ideologies defended. One defends the tribe through the political party and through the loyal press" (Ndembiyembe 1997, p. 47). We believe that a reorganization of media regulation in Cameroon, considering the results of our study, should more closely align with local socio-political realities. The idea is to reconsider ethical and deontological standards specific to the Cameroonian media environment.

**Funding:** This research constitutes part of the outcomes of my doctoral research, which received partial funding from ARES (ACADEMY OF RESEARCH AND HIGHER EDUCATION), a Belgian institution.

**Institutional Review Board Statement:** Not applicable.

**Informed Consent Statement:** Not applicable.

**Data Availability Statement:** The data presented in this study are available on request from the corresponding author.

**Conflicts of Interest:** The author declares no conflict of interest.

## Notes

[1] *Le Messenger* No.4347 Wednesday, 17 June 2015, p. 3

[2] *L'Œil du Sahel*, issue no. 692, Monday, 6 April 2015, p. 4.

[3] *Cameroon Tribune*, Tuesday, 2 September 2014, p. 6.

[4] At his accession to power, the Cameroonian president presented himself as the man of Renewal, and the creation of the Far North Region was his first act of territorial and administrative redistricting.

[5] *Cameroon Tribune* du mardi 9 Juin 2015, p. 9.

[6] *L'Oeil du Sahel* newspaper No. 666 of Monday 5 January 2015, p. 2.

[7] idem.

[8] *Cameroon Tribune* of Tuesday 9 June 2015, p. 9.

[9] *Cameroon Tribune*, Tuesday, 9 June 2015, p.9

[10] *Cameroon Tribune* of Tuesday 2 September 2014, p. 6.

[11] Interview conducted in *Cameroon Tribune* on 20 July 2017.

[12] *Le jour* no. 1854, Wednesday 21 January 2015, p. 5; *Le jour* no. 1870, Friday 13 February 2015, p. 3; *Le jour* no. 1897, Tuesday 24 March 2015, p. 2; *Le jour* no. 1888, Wednesday 11 March 2015, p. 3; *Le jour* no. 1903, Wednesday 1 April 2015, p. 7; *Le jour* no. 1931, Wednesday 13 May 2015, p. 4.

[13] *Le Messager* n°4298 Tuesday 7 April 2015 p. 4; *Le Messager* n°4274 Tuesday 3 March 2015, p. 2.

[14] *L'Oeil du Sahel* n°714 of Monday 22 June 2015, p. 11; *L'Oeil du Sahel* n°697 of Thursday 23 April 2015, p. 2-4; *L'Oeil du Sahel* n°693 of Thursday 09 April 2015 front cover; *L'Oeil du Sahel* n°703 of Friday 15 May 2015, p. 11; *L'Oeil du Sahel* n°704 of Monday 18 May 2015, p. 7; *L'Oeil du Sahel* n°692 of Monday 06 April 2015, p. 6; *L'Oeil du Sahel* n°715 of Thursday 25 June 2015, p. 11; *L'Oeil du Sahel* n°690 of Monday 30 March 2015, p. 9.

[15] *Cameroon Tribune* of Monday 30 March 2015, p. 10.

[16] *Ouest Echos* n°858 of 27 January to 2 February 2015, p. 10.

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
