# Peer review of "Digital Shifts and Ethno-Political Dynamics: Examining Event and Actor Designation in the Cameroon Boko Haram Terrorism Conflict through Print and Online Platforms"

_journalmedia, doi:10.3390/journalmedia5010024_

Round 1
Reviewer 1 Report
Comments and Suggestions for Authors
The content in Theoretical Approach and Methodology is repeated to some extent, I suggest reducing it; this part of the text is very extensive.
Repetition of paragraphs in lines 683-691
The manuscript lacks a clearly defined research problem (question), which will make the text more clear to the reader. Is this a question included in the abstract? If so, there is no clear answer to it in the conclusions.
Compared to the text as a whole, the conclusions are very laconic. Formulating the research question will enable the author(s) to expand the conclusions.
Comments on the Quality of English LanguagePlease read the text again, as it contains minor technical mistakes.
Author Response
REVIEWER 1
Thank you for your observations, which I welcome with humility. Below are the changes I have made to my text.
Yours sincerely,
- The content of the Theoretical Approach and Methodology section is somewhat repetitive. I suggest reducing it, as this portion of the text is particularly extensive.
To address this concern, I have confined my theoretical approach to the concept of designation (of actors and events), thereby reducing redundancy with the methodological section. The subsequent change is as follows:
- Old version
2. Theoretical Approach
Our approach revolves around the concept of designation, specifically focusing on event designation and actor designation as forms of informative framing. We delve into how these designations contribute to the unfolding of identity-related phenomena in both newspaper articles and online content within our corpus. In other words, we examine how designative choices shape the perception of the fight against terrorism in Cameroon, establishing the ethno-regional contours and stakes. Amossy (2005) discusses the sociality of discourse in this context, emphasizing that discourse is contingent upon its audience, to which the speaker must adapt (Perelman, 1989). "Seeking the sociality of the text is, above all, questioning who is speaking to whom, at what moment, from which place, under what circumstances, and in what space" (Amossy, 2005: 61).
Addressing the relationship of designation in this context aligns with the broader semiotic problematics of signs, particularly those that consider the study of meaning and its variations from the perspective of denomination or nomination (Kleiber, 1984, 2001, 2003; Frath, 2015, Petit, 2012), among others. Understanding the reality of the identity dynamics in Cameroon regarding the fight against terrorism through the lens of designation involves considering discursive phenomena that underlie the apprehension of signs by speaking subjects. To do so, it is crucial to distinguish what designation is not from what it is. Designation is not nomination, and certainly not denomination. To illustrate this distinction, we refer to the passage by Kleiber. According to him,
the two types of relations cannot be confused because, contrary to the sole relationship of designation, the relationship of denomination requires, that the relationship X (linguistic expression) — > x (things) has been established beforehand. Indeed, there is a relationship of denomination between X and x only if and only if there has been a prior act of denomination, i.e., the establishment of a referential link or a referential fixation, which can result from an effective act of denomination or simply from an associative habit, between the element x and the linguistic expression X. Such a requirement is not at all necessary for the relationship of designation. If I can only call something by its name if the thing has been named as such beforehand, I can designate, refer to, or point to something with an expression without that thing having been previously designated as such (Kleiber, 2001: 24).
Denomination is a kind of "label nomenclature, the inventory of which dictionaries compile, capturing the meaning conveyed by discourse" (Longhi, 2015: 5). Kleiber (2001) attributes several characteristics to it, including a pre-established fixed reference between the linguistic sign and the extralinguistic reality, a durable referential association, a preconstructed representational sense, an existential presupposition, etc. Nomination, on the other hand, categorizes a referent by placing it within a class of objects identified in the lexicon (Longhi, 2015: 6). According to Frath (2015: 39), it is a candidate to become a denomination.
Thus, designation involves referring to an extralinguistic object without any stable and fixed relationship between the linguistic sign and the referent having been previously established. The designator is therefore an unstabilized lexical form, a product of the creative imagination of the speaker. In journalistic work, it serves as a powerful framing tool, directing the perception of recipients through specific designation modalities (Calabrese 2009). This is because journalists select information to disseminate and often assign designations to the actors they discuss (Calabrese 2009, 2015). Therefore, how does designation operate in the newspaper articles and online content within our corpus? In what way does it account for the ethnic identity dynamics? On this matter, Van Ginneken, J. (1997) is quite prescriptive: "the production of meaning is intricately embedded in the activities of men and women in the institutions, or organization, and professions associated with their activities, and they produce and reproduce, create and recreate" (Ginneken, 1997: 18).
To better understand the act of designation in a newspaper and in the online content of our corpus within the scope of this study, it is essential to consider the mediation of journalists or webnotes, who update designative referents used to define ethnic identities, as well as the (identitarian or non-identitarian) relationship that connects them to their source. Identifiable identities in publications (online or otherwise) are only through words used to designate them. In other words, they are "the product of a journalistic naming protocol that has consequences in our perception" (Calabrese, 2008). Through the mediation of journalists and webnotes, they can control and neutralize the effects of these protocols (Boyomo, 2009; Tuchman, 1978). This is why designations in a newspaper or in online publications sometimes bear the marks of a struggle for the definition of the meaning to be given to current events. The act of designation is thus intimately linked to the editorial policy of a newspaper and in the online discussion forums of Cameroon. It is the arena of social struggles (Siblot, 1998). Its analysis reflects how the figure of the sender unfolds in the semiotic space of a given medium or platform, betraying intentions, goals, and the sought-after effect. This is made possible through how the contents of publications are handled in their enunciative structure. In our analysis, we specifically focus on event designators and actor designators within the studied printed and online publications.
2.1. Event designators
Event designators come into play when a new event emerges on the public stage, with the media (or webnotes) appropriating it through a process of designation (Calabrese, 2008). Through a series of operations, they seek to objectify it by defining its context and highlighting distinctive characteristics that facilitate its referencing (Ibidem).
Depending on the media's objectives or the webnote's intentions, the preferred characteristics in the designating term may include a date, a proper name, or even a "strict event name" (Calabrese, 2009) containing semes of eventuality (conflict against Boko Haram). Based on this, we refer to event designators as discourse elements used to describe, and sometimes define, situational contexts through their strong referential and semantic load, sometimes linked to a historical-media event (Calabrese, 2008, 2009). Their circulation through the media and continued presence in our daily conversations ensures their recognition throughout their duration in discourse (Calabrese, 2008). They have the capacity to store the coordinates of the event to which they refer (Calabrese, 2009). Alfred Schutz (1994) refers to this as a "social stock of shared knowledge." Simply mentioning their names is enough to trigger the recall of the events they suggest. "The media thus give a name to the event, which is supposed to describe it (even in its future and transformations) based on the dominant values in society. Some have a high degree of formatting (elections, heatwaves), others imply a certain interpretation of the context (a crisis, a war, riots), and a small number involve the public positioning of the speaker regarding a conflict (a genocide)" (Calabrese, 2009: 3).
Although the events suggested in our study by event designators did indeed have a physical reality, their verbalization is a socially constructed artifact. There are several types of event designators, including toponyms (which focus on the event's location as the main information coordinate), xenisms (aiding in the recall of toponymic coordinates through phonetic resonance and the exact naming of the event) and hemeronyms (event designators focusing on the date), etc.
In our research, we will delve further into hemeronyms. Morphosyntactically and semantically, the hemeronyms is a date that designates an event (Calabrese, 2008). Its eventuality load relies on an evocative context, and its formulation fulfills a need for linguistic economy aimed at producing a specific effect on the representation of the event (Calabrese, 2008). Pragmatically, the hemeronyms is capable of designating events limited in time, with a strong political component that interrupts the usual course of society. (...) it would thus have a very particular effect on the construction of the event, interrupting a natural temporality and establishing a transversal series of facts that, however, are not perceived homogeneously (Calabrese, 2008: 6). In other words, it is a form of event designator whose explanatory and predictive character is operational in the study of events of a political nature (Calabrese, 2008). In our view, this gives it sufficient relevance as an analytical instrument for examining the identity markers at play in the Cameroonian media space. The hemeronyms that captures our attention in the analyzed articles is the date of April 6. It interests us here for the narrative it suggests in the national opinion, especially during the conflict against Boko Haram.
Specifically in a context where the attacks of this terrorist group are perceived by a certain political elite belonging to the Béti ethnic conglomerate as "a rebellion in the North."
2.2. Actor designators
Actor designators play a crucial role in framing events (Bosk, Hilgartner, 1988: 58), as media outlets select information to disseminate and often assign designations to the actors they discuss. This task is carried out in reference to shared cognitive frameworks, professional protocols, and with the assistance of other social actors responsible for nomination and designation (e.g., the military, police) (Calabrese, 2015). A few months after Cameroon officially entered conflict with the Boko Haram terrorist group, different media outlets used distinct designations for this organization. Some national media referred to it as a "northern rebellion," a designation contested by others that earlier labeled it as "Boko Haram," depicting it as a "violent Islamist sect." In the former case, the designation attributes the aggression against Cameroon to a specific ethno-regional identity group, while the latter emphasizes an external aggression.
Hall (2007: 207) terms this situation "the struggle for positionalities," highlighting the challenges related to designating social actors in newspapers within an environment of ethno-regional identity competition for power benefits.
In summary, do the observed event and actor designators in print and online media reflect changes in the dynamics of the Cameroonian media sphere? How is designation relevant in depicting the fight against Boko Haram in Cameroonian press and Facebook forums? To answer these questions, the studied press and online content designation in this research are primarily understood as means of ethnic identification in the context of the Cameroonian socio-political environment governed by identity competition for power. It is also a strategy of territorial identification, with territoriality in the
Cameroonian socio-political field relating to a specific ethnic and linguistic identity (Abondo, 2022). This is why we consider the territoriality of the journal and that at play in the studied online content as a strategy of identity differentiation.
2.2.1. Press Designation in Cameroon as a Site of Power Relations
The theoretical foundations of the relationship between identity and media are rooted in the writings of Hall (2007: 203-214), particularly in "the new ethnicity." According to Hall, the relationship between identity and the discursive strategy of the media is grounded in ideological struggle, raising the issue of the "representation relationship" (Ibid: 204). Based on these approaches, the media constructs our understanding of ethnicity, suggesting how it should be perceived, understood, and the meanings it conveys. This practice relies on a significant effort in media representation. Hall emphasizes that "representation is only possible because enunciation always occurs within codes that have a history, occupying a position within the discursive forms of a particular time and space" (Hall, 2007: 209). In other words, "we all speak from a place, an experience, a particular history, a particular culture. [...] And in this sense, we are all ethnically located, and our ethnic identities are vital for the subjective feeling of who we are" (Ibid, 210).
In the Cameroonian context, this perspective views ethno-regional identities as a source of power in the realm of political competition. In other words, belonging to a specific ethno-regional group would be a political asset. This identity-focused view of political power is responsible for the transposition of ethnic identity reading into the Cameroonian media field. Due to the stakes associated with political competition, Cameroonian newspapers are labeled based on ethno-regional identities. Therefore, "the process of determining events and situations does not rely entirely on the media and involves a plurality of actors" (Arquembourg, 2005: 29).
This approach refers to press designations as a mediation site between diverse actors, stakeholders in a social reality (Garric, 2009). In the terminology of Ricœur (1983), this precisely concerns narrative operations in the mimetic process, allowing an understanding of how informational narratives achieve their goals (Arquembourg, 2005).
This is why we conceptualize designations as sites of embedding (Granovetter, 1985; D'Hont, Gérard, 2015, Le Velly, 2002; Laville, 2008, etc.) of power relations in a society. The representational approach of the media often results from the interweaving or entanglement of various power relations operating in society (Bobo, 1995; Hill Collins, 1990: 68-69; Hooks Bell 1992, 1996, Hall, 2007). These can be identity, religious, cultural, racial, and often economic. Therefore, Le Velly (2002) apprehends embedding as a sociology of market exchanges at the heart of social construction processes. Applied to the media field, the logic of embedding (Granovetter, 1985; D'Hont, Gérard, 2015, Le Velly, 2002; Laville, 2008, etc.) helps account for the fact that the interests of media actors and those of business and political environments form a system. This would result in event coverage based on the agreement model (Boltanski, Thévenot, 1991; Legavre, 2011).
In the Cameroonian sociopolitical context influenced by the philosophy of communitarian liberalism, this situation reveals various layers/surfaces of embedding and power relations that interconnect. Especially ethnic power relations, often associated with language, region, tribe, etc., further complicate matters from an analytical standpoint. It becomes challenging to ultimately identify what guides the editorial direction of newspapers and even online content. Our perspective also considers designations as editorial processes crystallizing current affairs within ethnic and territorial identity frameworks in a sociopolitical environment governed by competition over power issues. In other words, concerning the news studied in this research, designations of actors and events are primarily means of ethnic identification and territorialization in newspapers and online forums. What interests us in the analysis is how these identities, as articulated in print media and by new actors on social networks, shape popular representations of the studied social issue the fight against Boko Haram.
2.2.2. Press Designations and Their Relationships with Cameroonian Institutional Dynamics
Describing the media's relationship with the state under the former regime (1960-1982) in Cameroon, Thomas Atenga (2004) speaks of an environment where journalists had to primarily be patriots, even agents of development who unquestionably adhered to the President's discourse.
During this period, the journalist in Cameroon was "a vector of propaganda, sedative, calming, sleep-inducing information, in the name of national construction imperatives, maintaining public order, and safeguarding moral values. They were dedicated advocates of proselytism, brokers of the 'father of the nation's' words and activities" (Atenga, 2004: 38). In other words, the journalist's work was essentially framed within a "grammar of flattery" (Atenga, 2009: 72), where complicity prevailed, blending marketing, propaganda, and political advertising.
With the arrival of Paul Biya's new regime in 1982 and the liberalization of the press and associational movement in 1990, the Cameroonian state now had to learn to deal with a press that could criticize it and express opinions on certain decisions. Since the liberalization of the communication sector in the 1990s, establishing a newspaper in Cameroon requires very few administrative procedures. However, this shift was not automatic. It involved a struggle of nearly five years to transition from an authorization regime to a declaration regime, which was formalized in January 1996 with the amendment of the 1990 law, establishing freedom of communication. Until then, newspapers still faced the issue of prior censorship, which was abolished with the amendment of the same law.
In this context, the state and the media maintain conflictual and sometimes macabre relationships (Atenga, 2004), particularly in an authoritarian sociopolitical environment. The state apparatus, little accustomed to criticism, opened to a new era that witnessed an increased judicial crackdown on the media and journalists. Over 300 press trials were recorded in the early 1990s (Atenga, 2004: 5). "Combined attacks on freedom of expression (prior censorship, seizures, suspensions, bans, ransacking of newsrooms, physical assaults, journalist incarcerations, fiscal harassment, and various forms of intimidation) have far exceeded 2000" (Atenga, 2004: 5). "The announcement of a presidential visit abroad or the Chief's vacation destination could lead to a trial for endangering state security, while in democracies, the president's agenda and movements, as well as those of ministers and other personalities, are made public days in advance... A journalist can be sentenced by a court for substituting or holding administrative documents because he dared to present evidence of his allegations to the judge. The penalty is even heavier when he does not have them. In any case, he goes to trial condemned in advance" (Atenga, 2004: 10). This explains the limited maneuvering space in the use of event and actor designations.
In the age of digital social network (DSN) hegemony, the concept of designation experiences an extension, including in the Cameroonian context. As we have highlighted, this context is marked by ethno-regional identity interferences.
In fact, one of the changes brought about by the successive evolutions of the web 2.0 and 3.0 and the decrease in telephone communication costs is the new relationship with information created with Cameroonian media audiences. This has led to the fact that printed and traditional media in Cameroon are no longer the sole authorities in shaping public debate.
New agents have indeed entered defining the meaning of current news. We specifically think of the active participation of internet users/webnotes (Mercier, Pignard-Cheynel, 2014; Calabrese, Domingo, Pereira, 2015). They are now content producers in various digital environments. Their deployment brings about considerable changes in the way journalistic discourse is consumed (Calabrese, 2019). This phenomenon is growing given the numerous devices where an increasing number of readers get informed and express their opinions simultaneously. This leads to a migration of Cameroonian political events from printed newspapers to online discussion platforms. We are already talking about online political events (Calabrese, 2019). That is, an event that is "discussed in a digital environment and can, for that reason, adopt characteristics of web-native discourses" (Calabrese, 2019: 95). The Internet has also become a new political space (Castells, 2001) and a place for the reconfiguration of public space in Cameroon, where ethno-political identity conflicts also have a say. These platforms are extensions of conflicts in the classic media field and the birth of new conflicts around identities, power sharing, conservation, etc.
2.2.3. Press Designations as a Site for Ethno-Political Territorialization of the Cameroonian National Space
As mentioned earlier, the territorialization of the Cameroonian national space is strongly inclined towards the identity diversity (ethnic and linguistic) that characterizes the country. This initiative is part of the state's desire for "territorial differentiation" (Isabelle Pailliart, 2013: 278). This is a long-term process, initiated by both the state and the media, not only to designate the multiplication of territorial levels that mark the country's map but also, more fundamentally, the movement of differentiation in the social world (Pailliart, 2013). Thus, in Cameroon, the media are highly dependent on the state's territorialization policy. There are national, regional, departmental media, etc. They participate in controlling territorial differentiation and, by extension, social differentiation. It is in this aspect that we consider the territoriality of the newspaper in the analysis of designation. That is, as a movement of ethno-regional identity differentiation. An approach strongly challenged now with the advent of social networks, whose territoriality is global. Thus, they defy traditional obstacles of censorship, government repression, and economic pressures.
3. Methodology
This research combines critical analysis of press discourse with sociological analysis of journalistic production. Our approach aims to contextualize the ideology behind the designations of press articles and online content under study, particularly by examining their relationship to the mechanisms they result from, such as (for printed press) the territorial anchoring of the newspaper, its editorial line, its language of production, and (for both printed press and online content) the identity of the person speaking, or the community involved in the event.
The main objective is to establish a dialectical relationship between the discourse of newspapers, online content, and the sociological foundation of their production. The analysis of press discourse is based on the examination of 497 articles published between January 1 and June 30, 2015. They are drawn from 7 Cameroonian newspapers selected based on several criteria: the importance given to the identity (ethnic) question and the social claims it has generated. Other criteria such as language of publication, territorial coverage, and the origin of capital (public, private) were also considered. Moreover, these newspapers hold significant symbolic weight in structuring the public space in Cameroon despite the crisis prevailing in this field of activity. These include the bilingual government daily Cameroon Tribune (CT) (with a circulation between 8000 and 12000 copies), four private national dailies: two in French (Le Messager and Le Jour, each with a circulation between 3000 and 5000 copies), two in English (The Post and The Median, each with a circulation between 2500 and 5000 copies), and two regional newspapers (L’Œil du Sahel and Ouest Échos, each with a circulation between 2500 and 3000 copies).
The online corpus includes 450 written publications from the three Facebook forums: "Le Cameroun c'est le Cameroun (LCCLC)," "Le Cameroun pour tous (LCPT)," and "Le Cameroun que je veux voir." Our approach involved, initially, applying for membership in each of these online discussion groups. Once accepted as a member of these platforms, we conducted keyword searches in each group's search bar related to each of the news topics under study. Thus, in each of the Facebook groups, we initiated searches based on expressions such as "war against Boko Haram," "regional balance," and "Anglophone problem." The goal was to see all the publications produced in each of these groups on these specific topics. In addition to the search bar, the Facebook social network allows the delimitation of a specific period in relation to the launched search. Our observation in this regard is that the three selected groups produced nothing on our analyzed news topics in 2015, 2016, and 2017. They were created at the earliest in 2018 for the oldest, otherwise in 2019 for the others. Although social networks are now a prominent source of information in Cameroon, this reality is very recent. With a population of nearly 30 million, Facebook was still one of the favorite social networks for Cameroonians in the first quarter of 2022. By February 8 of the same year, there were more than 5 million account holders. Facebook ranked third among the most visited platforms in the country, behind YouTube in second position and Google in first position.
For the group "Le Cameroun, c'est le Cameroun (LCCLC)," which we studied, its oldest publications date back to 2018. We make this clarification because we identified four groups with the same name. After investigations with reliable sources, we learned that originally there was only one group with this name, created in 2015. The split into multiple groups would have occurred because the administrator of the original group had joined the MRC, a party whose candidate officially came second in the last presidential election of 2018. One of the groups resulting from this split is called "Le Cameroun c'est le Cameroun libre" ("Cameroon is a free Cameroon").
Concerning the groups "Le Cameroun pour tous (LCPT)" and "Le Cameroun que je veux voir," the oldest publications related to our research date back to 2019. On this basis, it was challenging for us to grasp the thoughts of internet users on our analytical themes within the study period from January to June 2015, which is our interval of investigation. We observed that several discussion groups of Cameroonians emerged between 2018 and 2019. Nevertheless, we chose to analyze this content as an extension of these news topics. The aim was to see if they are addressed through the same enunciative registers and if they carry the same imaginaries as in the print media, a few years earlier. The specific news topics that captured our attention are as follows: the fight against Boko Haram terrorism, the Anglophone crisis, and regional balance.
As for the sociology of journalistic production, it is based on 15 semi-directed interviews conducted with publishers, editorial managers, journalists who covered these events, officials from journalists' associations, and government officials dedicated to managing the media sector in Cameroon.
3.1. Data Processing and Analysis
To identify actor and event designations in our corpus, we plan to rely on what Gamson and Modigliani (1989) call "packages." These are elementary cores of meaning, which can be a word, a date, or an expression with a designation value related to ethno-regional identities, headlines, or simple statements. Our approach involves contextualizing the designations in our printed and online corpus in relation to the mechanisms they result from. This includes the territorial anchoring of the newspaper, editorial policy, language of publication, the identity of the media owner, and the person/community involved in the event. The objective is to establish a dialectical relationship between the discourse of newspapers and the sociological underpinning of their production. Subsequently, we will indicate how and in what manner the designation mode deployed in our online and print corpora reflects changes in the dynamics of the Cameroonian media sphere (censorship, government repression, economic pressures, etc.) concerning the emergence of new voices in digital discursive spaces.
To conduct this work, we initially compiled the main designations of actors and events associated with the fight against Boko Haram terrorism in our print corpus, especially in the four identified event sequences. Next, we sought to understand how they were discursively framed in different newspapers and the sociopolitical stakes they convey. We then examined the online content corpus to determine if the structure identified in the print media for the fight against Boko Haram was replicated and to identify the event and actor designations whenever they were used in this context. The observed differences will help understand the changes in the dynamics of the Cameroonian media sphere (censorship, government repression, economic pressures, etc.) with the emergence of new voices in digital discursive spaces. The quality of the designation observed in the studied forums will determine the actual change or mere complementarity between print media and online content. Both spheres of content publication exhibit a significant asymmetry in the rigor of state control. Online content offers greater flexibility and a wider margin of maneuver, as authorities may not have the means to act online as they do in print media.
3.2. Processing of Transcribed Interviews
The objective behind processing the transcribed interviews is to account for the role of ethno-regional identity interferences in the editorial attitude or professional behavior of media actors (journalists and publishers). As mentioned earlier, information here proves to be less a product of journalistic work and more a result of interactions among journalists, webnotes, and other actors, particularly those from business and political spheres (Champagne, 2006). These forces intertwine within the journalistic milieu, shaping its internal dynamics. Thus, the production of both print and online content is influenced by political realities. In this case, we consider ethno-regional identities within the context of the fight against terrorism.
Given the diverse profiles of our interviewees and the different positions from which each expressed themselves, maintaining a certain distance from the interview content became necessary. The goal was to conduct an analysis as detached as possible from the biases that characterize the statements of our interviewees. In such a context, our reflective approach and distancing in the analysis of interviews involved, each time, specifying the position of our interviewee, their background, possible conflicts of interest that could influence their speech in the context of this research, essentially placing them within the broader context of the mechanisms they are part of. From there, we sought to better understand the relevance of their statements to produce our analysis in relation to the phenomenon under study.
- New version
2. Theoretical Approach
Our approach revolves around the concept of designation, specifically focusing on event designation and actor designation as forms of informative framing. We delve into how these designations contribute to the unfolding of identity-related phenomena in both newspaper articles and online content within our corpus. In other words, we examine how designative choices shape the perception of the fight against terrorism in Cameroon, establishing the ethno-regional contours and stakes. Amossy (2005) discusses the sociality of discourse in this context, emphasizing that discourse is contingent upon its audience, to which the speaker must adapt (Perelman, 1989). Seeking the sociality of the text is, above all, questioning who is speaking to whom, at what moment, from which place, under what circumstances, and in what space (Amossy, 2005: 61).
Addressing the relationship of designation in this context aligns with the broader semiotic problematics of signs, particularly those that consider the study of meaning and its variations from the perspective of denomination or nomination (Kleiber, 1984, 2001, 2003; Frath, 2015, Petit, 2012), among others. Understanding the reality of the identity dynamics in Cameroon regarding the fight against terrorism through the lens of designation involves considering discursive phenomena that underlie the apprehension of signs by speaking subjects. To do so, it is crucial to distinguish what designation is not from what it is. Designation is not nomination, and certainly not denomination. To illustrate this distinction, we refer to the passage by Kleiber. According to him,
the two types of relations cannot be confused because, contrary to the sole relationship of designation, the relationship of denomination requires, that the relationship X (linguistic expression) — > x (things) has been established beforehand. Indeed, there is a relationship of denomination between X and x only if and only if there has been a prior act of denomination, i.e., the establishment of a referential link or a referential fixation, which can result from an effective act of denomination or simply from an associative habit, between the element x and the linguistic expression X. Such a requirement is not at all necessary for the relationship of designation. If I can only call something by its name if the thing has been named as such beforehand, I can designate, refer to, or point to something with an expression without that thing having been previously designated as such (Kleiber, 2001: 24).
Denomination is a kind of "label nomenclature, the inventory of which dictionaries compile, capturing the meaning conveyed by discourse" (Longhi, 2015: 5). Kleiber (2001) attributes several characteristics to it, including a pre-established fixed reference between the linguistic sign and the extralinguistic reality, a durable referential association, a preconstructed representational sense, an existential presupposition, etc. Nomination, on the other hand, categorizes a referent by placing it within a class of objects identified in the lexicon (Longhi, 2015: 6). According to Frath (2015: 39), it is a candidate to become a denomination.
Thus, designation involves referring to an extralinguistic object without any stable and fixed relationship between the linguistic sign and the referent having been previously established. The designator is therefore an unstabilized lexical form, a product of the creative imagination of the speaker. In journalistic work, it serves as a powerful framing tool, directing the perception of recipients through specific designation modalities (Calabrese 2009). This is because journalists select information to disseminate and often assign designations to the actors they discuss (Calabrese 2009, 2015). Therefore, how does designation operate in the newspaper articles and online content within our corpus? In what way does it account for the ethnic identity dynamics? On this matter, Van Ginneken, J. (1997) is quite prescriptive: "the production of meaning is intricately embedded in the activities of men and women in the institutions, or organization, and professions associated with their activities, and they produce and reproduce, create and recreate" (Ginneken, 1997: 18).
To better understand the act of designation in a newspaper and in the online content of our corpus within the scope of this study, it is essential to consider the mediation of journalists or webnotes, who update designative referents used to define ethnic identities, as well as the (identitarian or non-identitarian) relationship that connects them to their source. Identifiable identities in publications (online or otherwise) are only through words used to designate them. In other words, they are the product of a journalistic naming protocol that has consequences in our perception (Calabrese, 2008). Through the mediation of journalists and webnotes, they can control and neutralize the effects of these protocols (Boyomo, 2009; Tuchman, 1978). This is why designations in a newspaper or in online publications sometimes bear the marks of a struggle for the definition of the meaning to be given to current events. The act of designation is thus intimately linked to the editorial policy of a newspaper and in the online discussion forums of Cameroon. It is the arena of social struggles (Siblot, 1998). Its analysis reflects how the figure of the sender unfolds in the semiotic space of a given medium or platform, betraying intentions, goals, and the sought-after effect. This is made possible through how the contents of publications are handled in their enunciative structure. In our analysis, we specifically focus on event designators and actor designators within the studied printed and online publications.
Event designators
Event designators come into play when a new event emerges on the public stage, with the media (or webnotes) appropriating it through a process of designation (Calabrese, 2008). Through a series of operations, they seek to objectify it by defining its context and highlighting distinctive characteristics that facilitate its referencing (Ibidem).
Depending on the media's objectives or the webnote's intentions, the preferred characteristics in the designating term may include a date, a proper name, or even a strict event name (Calabrese, 2009) containing semes of eventuality (conflict against Boko Haram). Based on this, we refer to event designators as discourse elements used to describe, and sometimes define, situational contexts through their strong referential and semantic load, sometimes linked to a historical-media event (Calabrese, 2008, 2009). Their circulation through the media and continued presence in our daily conversations ensures their recognition throughout their duration in discourse (Calabrese, 2008). They have the capacity to store the coordinates of the event to which they refer (Calabrese, 2009). Alfred Schultz (1994) refers to this as a social stock of shared knowledge. Simply mentioning their names is enough to trigger the recall of the events they suggest. "The media thus give a name to the event, which is supposed to describe it (even in its future and transformations) based on the dominant values in society. Some have a high degree of formatting (elections, heatwaves), others imply a certain interpretation of the context (a crisis, a war, riots), and a small number involve the public positioning of the speaker regarding a conflict (a genocide)" (Calabrese, 2009: 3).
Although the events suggested in our study by event designators did indeed have a physical reality, their verbalization is a socially constructed artifact. There are several types of event designators, including toponyms (which focus on the event's location as the main information coordinate), xenisms (aiding in the recall of toponymic coordinates through phonetic resonance and the exact naming of the event) and hemeronyms (event designators focusing on the date), etc.
In our research, we will delve further into hemeronyms. Morphosyntactically and semantically, the hemeronyms is a date that designates an event (Calabrese, 2008). Its eventuality load relies on an evocative context, and its formulation fulfills a need for linguistic economy aimed at producing a specific effect on the representation of the event (Calabrese, 2008). Pragmatically, the hemeronyms is capable of designating events limited in time, with a strong political component that interrupts the usual course of society. (...) it would thus have a very particular effect on the construction of the event, interrupting a natural temporality and establishing a transversal series of facts that, however, are not perceived homogeneously (Calabrese, 2008: 6). In other words, it is a form of event designator whose explanatory and predictive character is operational in the study of events of a political nature (Calabrese, 2008). In our view, this gives it sufficient relevance as an analytical instrument for examining the identity markers at play in the Cameroonian media space. The hemeronyms that captures our attention in the analyzed articles is the date of April 6. It interests us here for the narrative it suggests in the national opinion, especially during the conflict against Boko Haram.
Specifically in a context where the attacks of this terrorist group are perceived by a certain political elite belonging to the Béti ethnic conglomerate as a rebellion in the North.
Actor designators
Actor designators play a crucial role in framing events (Bosk, Hilgartner, 1988: 58), as media outlets select information to disseminate and often assign designations to the actors they discuss. This task is carried out in reference to shared cognitive frameworks, professional protocols, and with the assistance of other social actors responsible for nomination and designation (e.g., the military, police) (Calabrese, 2015). A few months after Cameroon officially entered conflict with the Boko Haram terrorist group, different media outlets used distinct designations for this organization. Some national media referred to it as a northern rebellion, a designation contested by others that earlier labeled it as Boko Haram, depicting it as a violent Islamist sect. In the former case, the designation attributes the aggression against Cameroon to a specific ethno-regional identity group, while the latter emphasizes an external aggression.
Hall (2007: 207) terms this situation the struggle for positionalities, highlighting the challenges related to designating social actors in newspapers within an environment of ethno-regional identity competition for power benefits.
3. Methodology
This research combines critical analysis of press discourse with sociological analysis of journalistic production. Our approach aims to contextualize the ideology behind the designations of press articles and online content under study, particularly by examining their relationship to the mechanisms they result from, such as (for printed press) the territorial anchoring of the newspaper, its editorial line, its language of production, and (for both printed press and online content) the identity of the person speaking, or the community involved in the event.
The main objective is to establish a dialectical relationship between the discourse of newspapers, online content, and the sociological foundation of their production. The analysis of press discourse is based on the examination of 497 articles published between January 1 and June 30, 2015. They are drawn from 7 Cameroonian newspapers selected based on several criteria: the importance given to the identity (ethnic) question and the social claims it has generated. Other criteria such as language of publication, territorial coverage, and the origin of capital (public, private) were also considered. Moreover, these newspapers hold significant symbolic weight in structuring the public space in Cameroon despite the crisis prevailing in this field of activity. These include the bilingual government daily Cameroon Tribune (CT) (with a circulation between 8000 and 12000 copies), four private national dailies: two in French (Le Messager and Le Jour, each with a circulation between 3000 and 5000 copies), two in English (The Post and The Median, each with a circulation between 2500 and 5000 copies), and two regional newspapers (L’Œil du Sahel and Ouest Échos, each with a circulation between 2500 and 3000 copies).
The online corpus includes 450 written publications from the three Facebook forums: Le Cameroun c'est le Cameroun (LCCLC), Le Cameroun pour tous (LCPT), and Le Cameroun que je veux voir. Our approach involved, initially, applying for membership in each of these online discussion groups. Once accepted as a member of these platforms, we conducted keyword searches in each group's search bar related to each of the news topics under study. Thus, in each of the Facebook groups, we initiated searches based on expressions such as war against Boko Haram, regional balance, and Anglophone problem. The goal was to see all the publications produced in each of these groups on these specific topics. In addition to the search bar, the Facebook social network allows the delimitation of a specific period in relation to the launched search. Our observation in this regard is that the three selected groups produced nothing on our analyzed news topics in 2015, 2016, and 2017. They were created at the earliest in 2018 for the oldest, otherwise in 2019 for the others. Although social networks are now a prominent source of information in Cameroon, this reality is very recent. With a population of nearly 30 million, Facebook was still one of the favorite social networks for Cameroonians in the first quarter of 2022. By February 8 of the same year, there were more than 5 million account holders. Facebook ranked third among the most visited platforms in the country, behind YouTube in second position and Google in first position.
For the group Le Cameroun, c'est le Cameroun (LCCLC), which we studied, its oldest publications date back to 2018. We make this clarification because we identified four groups with the same name. After investigations with reliable sources, we learned that originally there was only one group with this name, created in 2015. The split into multiple groups would have occurred because the administrator of the original group had joined the MRC, a party whose candidate officially came second in the last presidential election of 2018. One of the groups resulting from this split is called Le Cameroun c'est le Cameroun libre (Cameroon is a free Cameroon).
Concerning the groups Le Cameroun pour tous (LCPT) and Le Cameroun que je veux voir, the oldest publications related to our research date back to 2019. On this basis, it was challenging for us to grasp the thoughts of internet users on our analytical themes within the study period from January to June 2015, which is our interval of investigation. We observed that several discussion groups of Cameroonians emerged between 2018 and 2019. Nevertheless, we chose to analyze this content as an extension of these news topics. The aim was to see if they are addressed through the same enunciative registers and if they carry the same imaginaries as in the print media, a few years earlier. The specific news topics that captured our attention are as follows: the fight against Boko Haram terrorism, the Anglophone crisis, and regional balance.
As for the sociology of journalistic production, it is based on 15 semi-directed interviews conducted with publishers, editorial managers, journalists who covered these events, officials from journalists' associations, and government officials dedicated to managing the media sector in Cameroon.
Table1. variables for the analysis of print and online contents.
|
Variable
corpus |
Territorial anchoring of the corpus |
Editorial line |
language of publication |
individuals/community involved in the event |
media owner (public/private) |
|
print corpus |
National, regional |
|
French, English |
Individuals and community |
Public and private |
|
Online corpus |
|
|
French, English |
Individuals and community |
Private |
Source: The author
3.1. Data Processing and Analysis
To identify actor and event designations in our corpus, we plan to rely on what Gamson and Modigliani (1989) call packages. These are elementary cores of meaning, which can be a word, a date, or an expression with a designation value related to ethno-regional identities, headlines, or simple statements. Our approach involves contextualizing the designations in our printed and online corpus in relation to the mechanisms they result from. This includes the territorial anchoring of the newspaper, editorial policy, language of publication, the identity of the media owner, and the person/community involved in the event. The objective is to establish a dialectical relationship between the discourse of newspapers and the sociological underpinning of their production. Subsequently, we will indicate how and in what manner the designation mode deployed in our online and print corpora reflects changes in the dynamics of the Cameroonian media sphere (censorship, government repression, economic pressures, etc.) concerning the emergence of new voices in digital discursive spaces.
To conduct this work, we initially compiled the main designations of actors and events associated with the fight against Boko Haram terrorism in our print corpus, especially in the four identified event sequences. Next, we sought to understand how they were discursively framed in different newspapers and the sociopolitical stakes they convey. We then examined the online content corpus to determine if the structure identified in the print media for the fight against Boko Haram was replicated and to identify the event and actor designations whenever they were used in this context. The observed differences will help understand the changes in the dynamics of the Cameroonian media sphere (censorship, government repression, economic pressures, etc.) with the emergence of new voices in digital discursive spaces. The quality of the designation observed in the studied forums will determine the actual change or mere complementarity between print media and online content. Both spheres of content publication exhibit a significant asymmetry in the rigor of state control. Online content offers greater flexibility and a wider margin of maneuver, as authorities may not have the means to act online as they do in print media.
3.2. Processing of Transcribed Interviews
The objective behind processing the transcribed interviews is to account for the role of ethno-regional identity interferences in the editorial attitude or professional behavior of media actors (journalists and publishers). As mentioned earlier, information here proves to be less a product of journalistic work and more a result of interactions among journalists, webnotes, and other actors, particularly those from business and political spheres (Champagne, 2006). These forces intertwine within the journalistic milieu, shaping its internal dynamics. Thus, the production of both print and online content is influenced by political realities. In this case, we consider ethno-regional identities within the context of the fight against terrorism.
Given the diverse profiles of our interviewees and the different positions from which each expressed themselves, maintaining a certain distance from the interview content became necessary. The goal was to conduct an analysis as detached as possible from the biases that characterize the statements of our interviewees. In such a context, our reflective approach and distancing in the analysis of interviews involved, each time, specifying the position of our interviewee, their background, possible conflicts of interest that could influence their speech in the context of this research, essentially placing them within the broader context of the mechanisms they are part of. From there, we sought to better understand the relevance of their statements to produce our analysis in relation to the phenomenon under study.
- There is redundancy in paragraphs 683-691.
To address this issue, I have removed one of the paragraphs. The resulting change is as follows:
- Old version
This reference is not at all insignificant, considering that the President of the National Assembly is indeed from the Far North region. This situation suggests various layers/surfaces of embedding and power relations that intersect with each other in this editorial office, particularly ethnic power relations often associated with language, region, tribe, etc.
This is not an insignificant allusion, as the President of the National Assembly is from the Far North region. This situation reveals the various layers/surfaces of embeddedness and power relations that are articulated in this drafting. These include ethnic power relations, often associated with language, region, tribe, etc.
- New version
This reference is not at all insignificant, considering that the President of the National Assembly is indeed from the Far North region. This situation suggests various layers/surfaces of embedding and power relations that intersect with each other in this editorial office, particularly ethnic power relations often associated with language, region, tribe, etc.
3- The manuscript lacks a clearly defined research problem (question), which will make the text more clear to the reader.
To address this issue, I have explicitly formulated the following research question:
- How does this technological shift affect the discourse, especially in the designation of events and actors in the reporting of Boko Haram terrorism in Cameroon, whether in traditional print media or on online platforms like Facebook?
- Do these designations in print media and Facebook discussion forums indicate shifts in the dynamics of the Cameroonian media sphere (censorship, government repression), resulting from the emergence of new voices in digital discursive spaces?
- In comparison to the rest of the text, the conclusions are very concise. The formulation of the research question would enable the author (or authors) to elaborate on the conclusions.
To address this issue, I have reformulated the research question as indicated above, and I have rewritten a conclusion that addresses these questions. The subsequent changes are as follows:
- Old conclusion
5. Conclusions
In sum, the analysis of the designation of events and actors in newspapers and online publications in our corpus around the fight against terrorism in Cameroon reveals changes in the dynamics of the media sphere (censorship, government repression and economic pressures etc.) Cameroon. In fact, with the establishment of social networks as the new place for information in Cameroon, there has been a transformation in the designation of events and actors in the fight against Boko Haram terrorism, given the ethnic identity issues at stake in the socio-political environment. The quality of the designation observed in the forums studied reflects a rise in hate speech in the ethno-political game in Cameroon. This phenomenon remained controllable with the action of state regulatory bodies. Unfortunately, the flexibility of online content combined with the absence of state control have given free rein to the rise of inter-ethnic discursive hatred in politics, according to our analytical data.
- New conclusion
5. Conclusions
In summary, the advent of social media in the dynamics of the Cameroonian media landscape has led to significant changes in journalism practices. It has spurred the emergence of new digital actors who now share the power to interpret news with journalists. This study observed this phenomenon through the analysis of event and actor designations in both print and online publications in our corpus, within the context of the fight against terrorism in Cameroon.
The findings indicate that the rise of social media as new information platforms in Cameroon has transformed the designation of events and actors in the fight against Boko Haram terrorism, intertwining it with the ethnic identity issues of the sociopolitical environment. While this reality was already present in print media, its migration to online content has metastasized into hate speech. This phenomenon, which was once controllable through the actions of state regulatory bodies, now reflects a transition from a press discourse environment regulated by state institutions to a realm where the state has limited control means. Consequently, full censorship and other forms of government repression are challenging to enforce.
Describing the relationship between the media and the state under the old regime (1960-1982) in Cameroon, Thomas Athenga (2004) speaks of an environment where journalists had to primarily be patriots or even development agents who unquestionably adhered to the President's discourse. Journalists in Cameroon during this period were "conveyors of propaganda, sedative, calming, and soporific information in the name of the imperatives of nation-building, maintenance of public order, and the preservation of good morals. They were dedicated proselytizers, brokers of the 'father of the nation's' speech and the activities of his administration" (Atenga, 2004: 38). In other words, journalists' work essentially conformed to a "grammar of flattery" (Atenga, 2009: 72), where connivance prevailed, blending marketing, propaganda, and political advertising.
With the arrival of Paul Biya's new regime in 1982 and the liberalization of the press and associative movement in 1990, the Cameroonian state must now learn to deal with a press that can criticize and express opinions on certain decisions. Since the liberalization of the communication sector in the 1990s, establishing a newspaper in Cameroon requires very few administrative procedures.
This did not happen spontaneously. It first involved a nearly five-year struggle to transition from the authorization regime to the declaration regime, which was formalized in January 1996 through the amendment of the 1990 law endorsing freedom of communication. Until then, newspapers still faced the issue of prior censorship, a problem that was also abolished with the amendment of the same law. In this context, the state and the media maintain conflictual, even macabre, relations (Atenga, 2004), within a sociopolitical environment described as authoritarian. Given the state apparatus's limited tolerance for criticism, a new era has emerged, witnessing an intensification of judicial repression against the media and journalists. Over 300 press trials were recorded in the early 1990s-2000s (Atenga, 2004: 5). "The cumulative infringements on freedom of expression (prior censorship, seizures, suspensions, bans, raids on editorial offices, physical assaults, imprisonment of journalists, fiscal harassment, and various forms of intimidation) have far exceeded 2000" (Atenga, 2004: 5). "The announcement of a presidential visit abroad or the Chief's vacation destination can lead to a trial for endangering state security, while in democracies, the president's agenda and movements, as well as those of ministers and other personalities, are made public days in advance… A journalist can be condemned by a court for the substitution or possession of administrative documents because he dared to present the judge with evidence of his allegations. The penalty is even more severe when he does not have them. In any case, he goes to trial already condemned" (Atenga, 2004: 10). This helps understand the limited maneuvering room in the use of event and actor designations.
However, with the advent of Web 3.0 in journalism, a new type of information relationship is now dominant through digital social networks (DSN). This has resulted in the infiltration of hate speech into the political-identity game in Cameroon. Our research results have sufficiently highlighted this through the uses of actor and event designations by web users. These designations are articulated with political-identity ethnic issues, demonstrating that media designation in Cameroon is a site of power relations.
The theoretical foundations of the relationship between identity and media are rooted in the writings of Hall (2007: 203-214), particularly in "New Ethnicity." According to Hall, the relationship between identity and media's discursive strategy is based on an ideological struggle, raising the question of "representation relationship" (Ibid: 204). Based on these approaches, the media shape our understanding of ethnicity, suggesting how it should be perceived, understood, and the meanings it conveys. This practice relies on significant effort in media representation. Hall emphasizes that "representation is only possible because enunciation always occurs in codes that have a history, occupying a position in the discursive forms of a specific time and space" (Hall, 2007: 209). In other words, "we all speak from a place, a particular experience, a specific history, a particular culture. [...] And in this sense, we are all ethnically located, and our ethnic identities are crucial to the subjective sense of who we are" (Ibid, 210).
In the Cameroonian context, this perspective views ethno-regional identities as a source of power in the realm of political competition. In other words, belonging to a specific ethno-regional group would be a political asset. This identity-centric vision of political power is responsible for the transposition of the reading of ethnic identity into the Cameroonian media field. Due to the stakes related to political competition, Cameroonian newspapers are labeled based on ethno-regional identities. Therefore, "the process of determining events and situations does not solely rely on the media and involves a plurality of actors" (Arquembourg, 2005: 29).
This approach refers to media designations as a mediation site among various actors, stakeholders in a social reality (Garric, 2009). In Ricœur's terminology (1983), these are precisely narrative operations in the mimetic process, allowing an understanding of how informative narratives achieve their objectives (Arquembourg, 2005).
Therefore, we conceptualize designations as sites of incorporation (Granovetter, 1985; D'Hont, Gérard, 2015, Le Velly, 2002; Laville, 2008, etc.) of power relations in society. The representational approach of the media often results from the intertwining or entanglement of various power relations operating in society (Bobo, 1995; Hill Collins, 1990: 68-69; Hooks Bell 1992, 1996, Hall, 2007). These can be identity-based, religious, cultural, racial, and often economic. Therefore, Le Velly (2002) apprehends incorporation as a sociology of market exchanges at the heart of social construction processes. Applied to the media domain, the logic of incorporation (Granovetter, 1985; D'Hont, Gérard, 2015, Le Velly, 2002; Laville, 2008, etc.) accounts for the fact that the interests of media actors and those of economic and political environments form a system. This would result in event coverage based on the agreement model (Boltanski, Thévenot, 1991; Legavre, 2011).
In the Cameroonian sociopolitical context influenced by the philosophy of communitarian liberalism (2018, 1986), this situation reveals various layers/surfaces of incorporation and power relations that intersect. Particularly, ethnic power relations, often associated with language, region, tribe, etc., further complicate things from an analytical perspective. It becomes challenging to ultimately identify what guides the editorial orientation of newspapers, including online content. Our perspective also considers designations as editorial processes crystallizing current events within frames of ethnic and territorial identity in a sociopolitical environment governed by competition on power issues. In other words, concerning the news studied in this research, designations of actors and events are primarily means of ethnic identification and territorialization in newspapers and online forums. To put it differently, the observed media designations in our corpus are also sites of ethno-political territorialization of the Cameroonian national space. Indeed, the territorialization of the Cameroonian national space is strongly oriented towards the identity diversity (ethnic and linguistic) characterizing the country.
This initiative is part of the state's desire for "territorial differentiation" (Isabelle Pailliart, 2013: 278). It is a long-term process, initiated by both the state and the media, aiming not only to designate the multiplication of territorial levels marking the country's map but also, more fundamentally, the movement of differentiation in the social world (Pailliart, 2013). In Cameroon, the media heavily relies on the state's policy of territorialization. There are national, regional, departmental, etc., media entities that contribute to mastering territorial differentiation and, by extension, social differentiation. It is in this aspect that we consider the territoriality of the newspaper in the analysis of the designation we conducted. That is, as a movement of differentiation of ethno-regional identity. An approach now strongly contested with the advent of social networks, whose territoriality is global. Thus, they challenge traditional obstacles such as censorship, government repression, and economic pressures, etc.
The way in which Social Media Networks (SMN) permeate the news studied in this research ultimately highlights the issue of media regulation in a multi-ethnic context in Africa. Ethnicity is a crucial variable around which power struggles are organized. Therefore, how should media regulation be reorganized in Cameroon in view of this challenge posed by SMN? Indeed, in addition to SMN, Cameroon is entrenched in a context of the end of a reign of over 40 years at the helm of the state, and where the press unfolds a "tribal war through intermediary newspapers [where] particularisms are thus magnified and defended. These are deviations that are justified, but when taken too far, can prove very dangerous. No longer are ideas or ideologies defended. One defends the tribe through the political party and through the loyal press" (Ndembiyembe, 1997: 47). We believe that a reorganization of media regulation in Cameroon, considering the results of our study, should more closely align with local socio-political realities. The idea is to reconsider ethical and deontological standards specific to the Cameroonian media environment.
Reviewer 2 Report
Comments and Suggestions for Authors
Dear authors,
Thank you very much to let me know reading your article that has a lot of potential and is very well written. However, I have some amendments to suggest you:
- On the on hand, some minor amendments should be considered:
o Abstract: the abstract will read better if it follows a more academic structure: objectives and research questions, methodology, and basic findings and conclusions.
o Introduction, line 90. The authors refer to the “disintermediation” process, but it should be noted that previously other authors have analyzed it in other contexts. Those previous works should be cited here.
o Introduction, lines 97-103. Do these questions refer to the research questions of this article? If these are RQ, please identify them correctly.
o Theoretical Approach, line 105. Does this section need a 2.1 sub-heading in order to introduce the content from line 105 to line 173? I recommend it in order to clarify the content.
o Theoretical approach, line 188. Be careful with direct quotations! They need to inform the page of the original document. Schultz (1994) direct quotation needs page referencing. Happens the same in line 217-218. Who is the author of this quotation? Check this issue through the rest of the document.
o Methodology. I suggest incorporating a table in order to situate the main variables that you use to select the main issues of the analysis, in the offline content and the online content. You explain correctly how you prepare the methodological skeleton, but a table or graph that can summarize the information will be helpful to clarify it.
- On the other hand, the main amendment of this article should be the inclusion, as part of the conclusions, melt within the results or in an individual section (5), a discussion. The article needs to deep into the intersection between the theoretical framework and the descriptive results it shows.
Hope these comments can help the authors to improve the article, which has a lot of potential and is needed to highlight the relationship between the media and one important conflict of Africa.
Author Response
REVIEWER 2
I am grateful for your kind words regarding my article. I have carefully considered each of your suggestions and made the necessary corrections. For each point discussed, please find below the detailed report of the modifications made. Your valuable input has greatly contributed to the enhancement of my work.
- Minor revisions :
- the abstract will read better if it follows a more academic structure: objectives and research questions, methodology, and basic findings and conclusions.
To address this concern, I have rewritten another summary. The subsequent changes are as follows:
- Old Abstract:
Abstract: The analysis delves into how the ethnic identity issue in Cameroon is addressed in the fight against Boko Haram terrorism at a time when new digital technologies have emerged, altering the practice of journalism. With the advent of social media and internet users, journalists have lost the monopoly on defining the meaning of news. Internet users operate with more freedom online, especially when outside the country, while journalists operate in an environment where their content are regulated by numerous state institutions. How this situation is reflecting in the modalities of speech, specifically the event and actor designations in the coverage of the fight against Boko Haram terrorism in Cameroon, both in print media and online platforms, such as Facebook? This study combines critical analysis of press discourse with sociological analysis of journalistic production. The analysis of press discourse is based on the examination of 497 articles published between January 1st and June 30, 2015. They are drawn from 7 Cameroonian newspapers. The online corpus includes 450 written publications from the three Facebook forums. The objective is to establish a dialectical relationship between the discourse of newspapers, online content, and the sociological foundation of their production. In sum, the quality of the designation observed in the forums studied reflects a rise in hate speech in the ethno-political game in Cameroon. This phenomenon remained controllable with the action of state regulatory bodies. Unfortunately, the flexibility of online content combined with the absence of state control have given free rein to the rise of inter-ethnic discursive hatred in politics.
Keywords: event and actor designation; Boko Haram terrorism; ethnic identity; print and online platforms; critical analysis of press discourse
- NewAbstract
Abstract: This study examines how the issue of ethnic identity is approached in Cameroon within the context of combating Boko Haram terrorism, considering the influence of the rise of social media on journalistic practices. The advent of these platforms has fundamentally altered the landscape of media coverage, challenging the traditional monopoly of journalists in shaping the narrative of news. How does this technological shift affect the discourse, especially in the designation of events and actors in the reporting of Boko Haram terrorism in Cameroon, whether in traditional print media or on online platforms like Facebook? Do these designations in print media and Facebook discussion forums indicate shifts in the dynamics of the Cameroonian media sphere (censorship, government repression etc.), resulting from the emergence of new voices in digital discursive spaces? This study employs a dual analysis, integrating a critical examination of media discourse with a sociological study of journalistic production. The scrutiny of media discourse is based on the investigation of 497 articles published between January 1st and June 30, 2015, sourced from 7 Cameroonian newspapers. The online corpus encompasses 450 written publications from three Facebook forums. We aim to establish a dialectical relationship between newspaper discourse, online content, and the sociological foundations shaping their production. The observed quality of designations in the studied forums unveils a surge in hate speech within the ethno-political landscape of Cameroon. While this phenomenon remained manageable through the intervention of state regulatory bodies in traditional media, the unrestrained nature of online content, coupled with the absence of state control, has facilitated the rise of inter-ethnic discursive hatred in politics. In conclusion, this study underscores the challenges stemming from the evolution of journalistic practices in a technological landscape and emphasizes the urgent need for regulatory frameworks to counteract the upswing in hate speech and interethnic tensions within political discourse.
Keywords: event and actor designation; Boko Haram terrorism; ethnic identity; print and online platforms; critical analysis of press discourse
- Introduction, line 90. The authors refer to the “disintermediation” process, but it should be noted that previously other authors have analyzed it in other contexts. Those previous works should be cited here.
To address this concern, I have rewritten another summary. The subsequent changes are as follows:
- Old version ligne 90
The arrival of new actors (webnotes/internet users) means that journalists are no longer the exclusive arbiters of the meaning of news.
- Old version
For Christensen C. M. (1997), Andreessen M. (2011), Tapscott, D., & Tapscott, A. (1994), Paul Maharg's (2016) terminology, this phenomenon is referred to as disintermediation. This involves a process whereby intermediaries in a supply chain are eliminated, typically through the digital re-engineering of processes and workflows. This process can lead to the dismantling of nearly all industries and professions and the restructuring of nearly every facet of customer-centric activities. In our context, this pertains to the relationship between journalists and their audiences.
- Introduction, lines 97-103. Do these questions refer to the research questions of this article? If these are RQ, please identify them correctly.
To address this concern, I have clarified the research questioning. The subsequent change is as follows:
- Old Inquiry/Questioning
Do these designations in print media and in the Facebook discussion forums under study reflect changes in the dynamics of the Cameroonian media sphere (such as censorship, government repression, economic pressures, etc.) with the emergence of new voices in digital discursive spaces? Has the rise of social media as new information platforms in Cameroon led to a transformation in the event and actor designations in the fight against Boko Haram, considering the ethnic identity issues in the sociopolitical environment?
- New Inquiry/Questioning
How does this technological shift affect the discourse, especially in the designation of events and actors in the reporting of Boko Haram terrorism in Cameroon, whether in traditional print media or on online platforms like Facebook?
Do these designations in print media and Facebook discussion forums indicate shifts in the dynamics of the Cameroonian media sphere (censorship, government repression etc.), resulting from the emergence of new voices in digital discursive spaces?
o Theoretical Approach, line 105. Does this section need a 2.1 sub-heading in order to introduce the content from line 105 to line 173? I recommend it in order to clarify the content.
To address this concern, I have rewritten the entire theoretical framework, focusing exclusively on the concept of designation (of actors and events). The subsequent change is as follows:
- Old Theoretical Framework
2. Theoretical Approach
Our approach revolves around the concept of designation, specifically focusing on event designation and actor designation as forms of informative framing. We delve into how these designations contribute to the unfolding of identity-related phenomena in both newspaper articles and online content within our corpus. In other words, we examine how designative choices shape the perception of the fight against terrorism in Cameroon, establishing the ethno-regional contours and stakes. Amossy (2005) discusses the sociality of discourse in this context, emphasizing that discourse is contingent upon its audience, to which the speaker must adapt (Perelman, 1989). "Seeking the sociality of the text is, above all, questioning who is speaking to whom, at what moment, from which place, under what circumstances, and in what space" (Amossy, 2005: 61).
Addressing the relationship of designation in this context aligns with the broader semiotic problematics of signs, particularly those that consider the study of meaning and its variations from the perspective of denomination or nomination (Kleiber, 1984, 2001, 2003; Frath, 2015, Petit, 2012), among others. Understanding the reality of the identity dynamics in Cameroon regarding the fight against terrorism through the lens of designation involves considering discursive phenomena that underlie the apprehension of signs by speaking subjects. To do so, it is crucial to distinguish what designation is not from what it is. Designation is not nomination, and certainly not denomination. To illustrate this distinction, we refer to the passage by Kleiber. According to him,
the two types of relations cannot be confused because, contrary to the sole relationship of designation, the relationship of denomination requires, that the relationship X (linguistic expression) — > x (things) has been established beforehand. Indeed, there is a relationship of denomination between X and x only if and only if there has been a prior act of denomination, i.e., the establishment of a referential link or a referential fixation, which can result from an effective act of denomination or simply from an associative habit, between the element x and the linguistic expression X. Such a requirement is not at all necessary for the relationship of designation. If I can only call something by its name if the thing has been named as such beforehand, I can designate, refer to, or point to something with an expression without that thing having been previously designated as such (Kleiber, 2001: 24).
Denomination is a kind of "label nomenclature, the inventory of which dictionaries compile, capturing the meaning conveyed by discourse" (Longhi, 2015: 5). Kleiber (2001) attributes several characteristics to it, including a pre-established fixed reference between the linguistic sign and the extralinguistic reality, a durable referential association, a preconstructed representational sense, an existential presupposition, etc. Nomination, on the other hand, categorizes a referent by placing it within a class of objects identified in the lexicon (Longhi, 2015: 6). According to Frath (2015: 39), it is a candidate to become a denomination.
Thus, designation involves referring to an extralinguistic object without any stable and fixed relationship between the linguistic sign and the referent having been previously established. The designator is therefore an unstabilized lexical form, a product of the creative imagination of the speaker. In journalistic work, it serves as a powerful framing tool, directing the perception of recipients through specific designation modalities (Calabrese 2009). This is because journalists select information to disseminate and often assign designations to the actors they discuss (Calabrese 2009, 2015). Therefore, how does designation operate in the newspaper articles and online content within our corpus? In what way does it account for the ethnic identity dynamics? On this matter, Van Ginneken, J. (1997) is quite prescriptive: "the production of meaning is intricately embedded in the activities of men and women in the institutions, or organization, and professions associated with their activities, and they produce and reproduce, create and recreate" (Ginneken, 1997: 18).
To better understand the act of designation in a newspaper and in the online content of our corpus within the scope of this study, it is essential to consider the mediation of journalists or webnotes, who update designative referents used to define ethnic identities, as well as the (identitarian or non-identitarian) relationship that connects them to their source. Identifiable identities in publications (online or otherwise) are only through words used to designate them. In other words, they are "the product of a journalistic naming protocol that has consequences in our perception" (Calabrese, 2008). Through the mediation of journalists and webnotes, they can control and neutralize the effects of these protocols (Boyomo, 2009; Tuchman, 1978). This is why designations in a newspaper or in online publications sometimes bear the marks of a struggle for the definition of the meaning to be given to current events. The act of designation is thus intimately linked to the editorial policy of a newspaper and in the online discussion forums of Cameroon. It is the arena of social struggles (Siblot, 1998). Its analysis reflects how the figure of the sender unfolds in the semiotic space of a given medium or platform, betraying intentions, goals, and the sought-after effect. This is made possible through how the contents of publications are handled in their enunciative structure. In our analysis, we specifically focus on event designators and actor designators within the studied printed and online publications.
2.1. Event designators
Event designators come into play when a new event emerges on the public stage, with the media (or webnotes) appropriating it through a process of designation (Calabrese, 2008). Through a series of operations, they seek to objectify it by defining its context and highlighting distinctive characteristics that facilitate its referencing (Ibidem).
Depending on the media's objectives or the webnote's intentions, the preferred characteristics in the designating term may include a date, a proper name, or even a "strict event name" (Calabrese, 2009) containing semes of eventuality (conflict against Boko Haram). Based on this, we refer to event designators as discourse elements used to describe, and sometimes define, situational contexts through their strong referential and semantic load, sometimes linked to a historical-media event (Calabrese, 2008, 2009). Their circulation through the media and continued presence in our daily conversations ensures their recognition throughout their duration in discourse (Calabrese, 2008). They have the capacity to store the coordinates of the event to which they refer (Calabrese, 2009). Alfred Schutz (1994) refers to this as a "social stock of shared knowledge." Simply mentioning their names is enough to trigger the recall of the events they suggest. "The media thus give a name to the event, which is supposed to describe it (even in its future and transformations) based on the dominant values in society. Some have a high degree of formatting (elections, heatwaves), others imply a certain interpretation of the context (a crisis, a war, riots), and a small number involve the public positioning of the speaker regarding a conflict (a genocide)" (Calabrese, 2009: 3).
Although the events suggested in our study by event designators did indeed have a physical reality, their verbalization is a socially constructed artifact. There are several types of event designators, including toponyms (which focus on the event's location as the main information coordinate), xenisms (aiding in the recall of toponymic coordinates through phonetic resonance and the exact naming of the event) and hemeronyms (event designators focusing on the date), etc.
In our research, we will delve further into hemeronyms. Morphosyntactically and semantically, the hemeronyms is a date that designates an event (Calabrese, 2008). Its eventuality load relies on an evocative context, and its formulation fulfills a need for linguistic economy aimed at producing a specific effect on the representation of the event (Calabrese, 2008). Pragmatically, the hemeronyms is capable of designating events limited in time, with a strong political component that interrupts the usual course of society. (...) it would thus have a very particular effect on the construction of the event, interrupting a natural temporality and establishing a transversal series of facts that, however, are not perceived homogeneously (Calabrese, 2008: 6). In other words, it is a form of event designator whose explanatory and predictive character is operational in the study of events of a political nature (Calabrese, 2008). In our view, this gives it sufficient relevance as an analytical instrument for examining the identity markers at play in the Cameroonian media space. The hemeronyms that captures our attention in the analyzed articles is the date of April 6. It interests us here for the narrative it suggests in the national opinion, especially during the conflict against Boko Haram.
Specifically in a context where the attacks of this terrorist group are perceived by a certain political elite belonging to the Béti ethnic conglomerate as "a rebellion in the North."
2.2. Actor designators
Actor designators play a crucial role in framing events (Bosk, Hilgartner, 1988: 58), as media outlets select information to disseminate and often assign designations to the actors they discuss. This task is carried out in reference to shared cognitive frameworks, professional protocols, and with the assistance of other social actors responsible for nomination and designation (e.g., the military, police) (Calabrese, 2015). A few months after Cameroon officially entered conflict with the Boko Haram terrorist group, different media outlets used distinct designations for this organization. Some national media referred to it as a "northern rebellion," a designation contested by others that earlier labeled it as "Boko Haram," depicting it as a "violent Islamist sect." In the former case, the designation attributes the aggression against Cameroon to a specific ethno-regional identity group, while the latter emphasizes an external aggression.
Hall (2007: 207) terms this situation "the struggle for positionalities," highlighting the challenges related to designating social actors in newspapers within an environment of ethno-regional identity competition for power benefits.
In summary, do the observed event and actor designators in print and online media reflect changes in the dynamics of the Cameroonian media sphere? How is designation relevant in depicting the fight against Boko Haram in Cameroonian press and Facebook forums? To answer these questions, the studied press and online content designation in this research are primarily understood as means of ethnic identification in the context of the Cameroonian socio-political environment governed by identity competition for power. It is also a strategy of territorial identification, with territoriality in the
Cameroonian socio-political field relating to a specific ethnic and linguistic identity (Abondo, 2022). This is why we consider the territoriality of the journal and that at play in the studied online content as a strategy of identity differentiation.
2.2.1. Press Designation in Cameroon as a Site of Power Relations
The theoretical foundations of the relationship between identity and media are rooted in the writings of Hall (2007: 203-214), particularly in "the new ethnicity." According to Hall, the relationship between identity and the discursive strategy of the media is grounded in ideological struggle, raising the issue of the "representation relationship" (Ibid: 204). Based on these approaches, the media constructs our understanding of ethnicity, suggesting how it should be perceived, understood, and the meanings it conveys. This practice relies on a significant effort in media representation. Hall emphasizes that "representation is only possible because enunciation always occurs within codes that have a history, occupying a position within the discursive forms of a particular time and space" (Hall, 2007: 209). In other words, "we all speak from a place, an experience, a particular history, a particular culture. [...] And in this sense, we are all ethnically located, and our ethnic identities are vital for the subjective feeling of who we are" (Ibid, 210).
In the Cameroonian context, this perspective views ethno-regional identities as a source of power in the realm of political competition. In other words, belonging to a specific ethno-regional group would be a political asset. This identity-focused view of political power is responsible for the transposition of ethnic identity reading into the Cameroonian media field. Due to the stakes associated with political competition, Cameroonian newspapers are labeled based on ethno-regional identities. Therefore, "the process of determining events and situations does not rely entirely on the media and involves a plurality of actors" (Arquembourg, 2005: 29).
This approach refers to press designations as a mediation site between diverse actors, stakeholders in a social reality (Garric, 2009). In the terminology of Ricœur (1983), this precisely concerns narrative operations in the mimetic process, allowing an understanding of how informational narratives achieve their goals (Arquembourg, 2005).
This is why we conceptualize designations as sites of embedding (Granovetter, 1985; D'Hont, Gérard, 2015, Le Velly, 2002; Laville, 2008, etc.) of power relations in a society. The representational approach of the media often results from the interweaving or entanglement of various power relations operating in society (Bobo, 1995; Hill Collins, 1990: 68-69; Hooks Bell 1992, 1996, Hall, 2007). These can be identity, religious, cultural, racial, and often economic. Therefore, Le Velly (2002) apprehends embedding as a sociology of market exchanges at the heart of social construction processes. Applied to the media field, the logic of embedding (Granovetter, 1985; D'Hont, Gérard, 2015, Le Velly, 2002; Laville, 2008, etc.) helps account for the fact that the interests of media actors and those of business and political environments form a system. This would result in event coverage based on the agreement model (Boltanski, Thévenot, 1991; Legavre, 2011).
In the Cameroonian sociopolitical context influenced by the philosophy of communitarian liberalism, this situation reveals various layers/surfaces of embedding and power relations that interconnect. Especially ethnic power relations, often associated with language, region, tribe, etc., further complicate matters from an analytical standpoint. It becomes challenging to ultimately identify what guides the editorial direction of newspapers and even online content. Our perspective also considers designations as editorial processes crystallizing current affairs within ethnic and territorial identity frameworks in a sociopolitical environment governed by competition over power issues. In other words, concerning the news studied in this research, designations of actors and events are primarily means of ethnic identification and territorialization in newspapers and online forums. What interests us in the analysis is how these identities, as articulated in print media and by new actors on social networks, shape popular representations of the studied social issue the fight against Boko Haram.
2.2.2. Press Designations and Their Relationships with Cameroonian Institutional Dynamics
Describing the media's relationship with the state under the former regime (1960-1982) in Cameroon, Thomas Atenga (2004) speaks of an environment where journalists had to primarily be patriots, even agents of development who unquestionably adhered to the President's discourse.
During this period, the journalist in Cameroon was "a vector of propaganda, sedative, calming, sleep-inducing information, in the name of national construction imperatives, maintaining public order, and safeguarding moral values. They were dedicated advocates of proselytism, brokers of the 'father of the nation's' words and activities" (Atenga, 2004: 38). In other words, the journalist's work was essentially framed within a "grammar of flattery" (Atenga, 2009: 72), where complicity prevailed, blending marketing, propaganda, and political advertising.
With the arrival of Paul Biya's new regime in 1982 and the liberalization of the press and associational movement in 1990, the Cameroonian state now had to learn to deal with a press that could criticize it and express opinions on certain decisions. Since the liberalization of the communication sector in the 1990s, establishing a newspaper in Cameroon requires very few administrative procedures. However, this shift was not automatic. It involved a struggle of nearly five years to transition from an authorization regime to a declaration regime, which was formalized in January 1996 with the amendment of the 1990 law, establishing freedom of communication. Until then, newspapers still faced the issue of prior censorship, which was abolished with the amendment of the same law.
In this context, the state and the media maintain conflictual and sometimes macabre relationships (Atenga, 2004), particularly in an authoritarian sociopolitical environment. The state apparatus, little accustomed to criticism, opened to a new era that witnessed an increased judicial crackdown on the media and journalists. Over 300 press trials were recorded in the early 1990s (Atenga, 2004: 5). "Combined attacks on freedom of expression (prior censorship, seizures, suspensions, bans, ransacking of newsrooms, physical assaults, journalist incarcerations, fiscal harassment, and various forms of intimidation) have far exceeded 2000" (Atenga, 2004: 5). "The announcement of a presidential visit abroad or the Chief's vacation destination could lead to a trial for endangering state security, while in democracies, the president's agenda and movements, as well as those of ministers and other personalities, are made public days in advance... A journalist can be sentenced by a court for substituting or holding administrative documents because he dared to present evidence of his allegations to the judge. The penalty is even heavier when he does not have them. In any case, he goes to trial condemned in advance" (Atenga, 2004: 10). This explains the limited maneuvering space in the use of event and actor designations.
In the age of digital social network (DSN) hegemony, the concept of designation experiences an extension, including in the Cameroonian context. As we have highlighted, this context is marked by ethno-regional identity interferences.
In fact, one of the changes brought about by the successive evolutions of the web 2.0 and 3.0 and the decrease in telephone communication costs is the new relationship with information created with Cameroonian media audiences. This has led to the fact that printed and traditional media in Cameroon are no longer the sole authorities in shaping public debate.
New agents have indeed entered defining the meaning of current news. We specifically think of the active participation of internet users/webnotes (Mercier, Pignard-Cheynel, 2014; Calabrese, Domingo, Pereira, 2015). They are now content producers in various digital environments. Their deployment brings about considerable changes in the way journalistic discourse is consumed (Calabrese, 2019). This phenomenon is growing given the numerous devices where an increasing number of readers get informed and express their opinions simultaneously. This leads to a migration of Cameroonian political events from printed newspapers to online discussion platforms. We are already talking about online political events (Calabrese, 2019). That is, an event that is "discussed in a digital environment and can, for that reason, adopt characteristics of web-native discourses" (Calabrese, 2019: 95). The Internet has also become a new political space (Castells, 2001) and a place for the reconfiguration of public space in Cameroon, where ethno-political identity conflicts also have a say. These platforms are extensions of conflicts in the classic media field and the birth of new conflicts around identities, power sharing, conservation, etc.
2.2.3. Press Designations as a Site for Ethno-Political Territorialization of the Cameroonian National Space
As mentioned earlier, the territorialization of the Cameroonian national space is strongly inclined towards the identity diversity (ethnic and linguistic) that characterizes the country. This initiative is part of the state's desire for "territorial differentiation" (Isabelle Pailliart, 2013: 278). This is a long-term process, initiated by both the state and the media, not only to designate the multiplication of territorial levels that mark the country's map but also, more fundamentally, the movement of differentiation in the social world (Pailliart, 2013). Thus, in Cameroon, the media are highly dependent on the state's territorialization policy. There are national, regional, departmental media, etc. They participate in controlling territorial differentiation and, by extension, social differentiation. It is in this aspect that we consider the territoriality of the newspaper in the analysis of designation. That is, as a movement of ethno-regional identity differentiation. An approach strongly challenged now with the advent of social networks, whose territoriality is global. Thus, they defy traditional obstacles of censorship, government repression, and economic pressures.
- New Theoretical Framework
2. Theoretical Approach
Our approach revolves around the concept of designation, specifically focusing on event designation and actor designation as forms of informative framing. We delve into how these designations contribute to the unfolding of identity-related phenomena in both newspaper articles and online content within our corpus. In other words, we examine how designative choices shape the perception of the fight against terrorism in Cameroon, establishing the ethno-regional contours and stakes. Amossy (2005) discusses the sociality of discourse in this context, emphasizing that discourse is contingent upon its audience, to which the speaker must adapt (Perelman, 1989). Seeking the sociality of the text is, above all, questioning who is speaking to whom, at what moment, from which place, under what circumstances, and in what space (Amossy, 2005: 61).
Addressing the relationship of designation in this context aligns with the broader semiotic problematics of signs, particularly those that consider the study of meaning and its variations from the perspective of denomination or nomination (Kleiber, 1984, 2001, 2003; Frath, 2015, Petit, 2012), among others. Understanding the reality of the identity dynamics in Cameroon regarding the fight against terrorism through the lens of designation involves considering discursive phenomena that underlie the apprehension of signs by speaking subjects. To do so, it is crucial to distinguish what designation is not from what it is. Designation is not nomination, and certainly not denomination. To illustrate this distinction, we refer to the passage by Kleiber. According to him,
the two types of relations cannot be confused because, contrary to the sole relationship of designation, the relationship of denomination requires, that the relationship X (linguistic expression) — > x (things) has been established beforehand. Indeed, there is a relationship of denomination between X and x only if and only if there has been a prior act of denomination, i.e., the establishment of a referential link or a referential fixation, which can result from an effective act of denomination or simply from an associative habit, between the element x and the linguistic expression X. Such a requirement is not at all necessary for the relationship of designation. If I can only call something by its name if the thing has been named as such beforehand, I can designate, refer to, or point to something with an expression without that thing having been previously designated as such (Kleiber, 2001: 24).
Denomination is a kind of "label nomenclature, the inventory of which dictionaries compile, capturing the meaning conveyed by discourse" (Longhi, 2015: 5). Kleiber (2001) attributes several characteristics to it, including a pre-established fixed reference between the linguistic sign and the extralinguistic reality, a durable referential association, a preconstructed representational sense, an existential presupposition, etc. Nomination, on the other hand, categorizes a referent by placing it within a class of objects identified in the lexicon (Longhi, 2015: 6). According to Frath (2015: 39), it is a candidate to become a denomination.
Thus, designation involves referring to an extralinguistic object without any stable and fixed relationship between the linguistic sign and the referent having been previously established. The designator is therefore an unstabilized lexical form, a product of the creative imagination of the speaker. In journalistic work, it serves as a powerful framing tool, directing the perception of recipients through specific designation modalities (Calabrese 2009). This is because journalists select information to disseminate and often assign designations to the actors they discuss (Calabrese 2009, 2015). Therefore, how does designation operate in the newspaper articles and online content within our corpus? In what way does it account for the ethnic identity dynamics? On this matter, Van Ginneken, J. (1997) is quite prescriptive: "the production of meaning is intricately embedded in the activities of men and women in the institutions, or organization, and professions associated with their activities, and they produce and reproduce, create and recreate" (Ginneken, 1997: 18).
To better understand the act of designation in a newspaper and in the online content of our corpus within the scope of this study, it is essential to consider the mediation of journalists or webnotes, who update designative referents used to define ethnic identities, as well as the (identitarian or non-identitarian) relationship that connects them to their source. Identifiable identities in publications (online or otherwise) are only through words used to designate them. In other words, they are the product of a journalistic naming protocol that has consequences in our perception (Calabrese, 2008). Through the mediation of journalists and webnotes, they can control and neutralize the effects of these protocols (Boyomo, 2009; Tuchman, 1978). This is why designations in a newspaper or in online publications sometimes bear the marks of a struggle for the definition of the meaning to be given to current events. The act of designation is thus intimately linked to the editorial policy of a newspaper and in the online discussion forums of Cameroon. It is the arena of social struggles (Siblot, 1998). Its analysis reflects how the figure of the sender unfolds in the semiotic space of a given medium or platform, betraying intentions, goals, and the sought-after effect. This is made possible through how the contents of publications are handled in their enunciative structure. In our analysis, we specifically focus on event designators and actor designators within the studied printed and online publications.
Event designators
Event designators come into play when a new event emerges on the public stage, with the media (or webnotes) appropriating it through a process of designation (Calabrese, 2008). Through a series of operations, they seek to objectify it by defining its context and highlighting distinctive characteristics that facilitate its referencing (Ibidem).
Depending on the media's objectives or the webnote's intentions, the preferred characteristics in the designating term may include a date, a proper name, or even a strict event name (Calabrese, 2009) containing semes of eventuality (conflict against Boko Haram). Based on this, we refer to event designators as discourse elements used to describe, and sometimes define, situational contexts through their strong referential and semantic load, sometimes linked to a historical-media event (Calabrese, 2008, 2009). Their circulation through the media and continued presence in our daily conversations ensures their recognition throughout their duration in discourse (Calabrese, 2008). They have the capacity to store the coordinates of the event to which they refer (Calabrese, 2009). Alfred Schultz (1994) refers to this as a social stock of shared knowledge. Simply mentioning their names is enough to trigger the recall of the events they suggest. "The media thus give a name to the event, which is supposed to describe it (even in its future and transformations) based on the dominant values in society. Some have a high degree of formatting (elections, heatwaves), others imply a certain interpretation of the context (a crisis, a war, riots), and a small number involve the public positioning of the speaker regarding a conflict (a genocide)" (Calabrese, 2009: 3).
Although the events suggested in our study by event designators did indeed have a physical reality, their verbalization is a socially constructed artifact. There are several types of event designators, including toponyms (which focus on the event's location as the main information coordinate), xenisms (aiding in the recall of toponymic coordinates through phonetic resonance and the exact naming of the event) and hemeronyms (event designators focusing on the date), etc.
In our research, we will delve further into hemeronyms. Morphosyntactically and semantically, the hemeronyms is a date that designates an event (Calabrese, 2008). Its eventuality load relies on an evocative context, and its formulation fulfills a need for linguistic economy aimed at producing a specific effect on the representation of the event (Calabrese, 2008). Pragmatically, the hemeronyms is capable of designating events limited in time, with a strong political component that interrupts the usual course of society. (...) it would thus have a very particular effect on the construction of the event, interrupting a natural temporality and establishing a transversal series of facts that, however, are not perceived homogeneously (Calabrese, 2008: 6). In other words, it is a form of event designator whose explanatory and predictive character is operational in the study of events of a political nature (Calabrese, 2008). In our view, this gives it sufficient relevance as an analytical instrument for examining the identity markers at play in the Cameroonian media space. The hemeronyms that captures our attention in the analyzed articles is the date of April 6. It interests us here for the narrative it suggests in the national opinion, especially during the conflict against Boko Haram.
Specifically in a context where the attacks of this terrorist group are perceived by a certain political elite belonging to the Béti ethnic conglomerate as a rebellion in the North.
Actor designators
Actor designators play a crucial role in framing events (Bosk, Hilgartner, 1988: 58), as media outlets select information to disseminate and often assign designations to the actors they discuss. This task is carried out in reference to shared cognitive frameworks, professional protocols, and with the assistance of other social actors responsible for nomination and designation (e.g., the military, police) (Calabrese, 2015). A few months after Cameroon officially entered conflict with the Boko Haram terrorist group, different media outlets used distinct designations for this organization. Some national media referred to it as a northern rebellion, a designation contested by others that earlier labeled it as Boko Haram, depicting it as a violent Islamist sect. In the former case, the designation attributes the aggression against Cameroon to a specific ethno-regional identity group, while the latter emphasizes an external aggression.
Hall (2007: 207) terms this situation the struggle for positionalities, highlighting the challenges related to designating social actors in newspapers within an environment of ethno-regional identity competition for power benefits.
o Theoretical approach, line 188. Be careful with direct quotations! They need to inform the page of the original document. Schultz (1994) direct quotation needs page referencing. Happens the same in line 217-218. Who is the author of this quotation? Check this issue through the rest of the document.
Pour répondre à cette préoccupation et à toutes les autres du même type dans l’article j’ai souvent enlevé les guillemets par fois j’ai rajouté la page. Dans le cas des deux exemple cité ci-dessus j’ai enlevé les guillemets.
- Methodology: I suggest incorporating a table in order to situate the main variables that you use to select the main issues of the analysis, in the offline content and the online content. You explain correctly how you prepare the methodological skeleton, but a table or graph that can summarize the information will be helpful to clarify it.
To address this concern, I have created a table of analysis variables for each type of studied corpus below:
Table1. variables for the analysis of print and online contents.
|
Variable
corpus |
Territorial anchoring of the corpus |
Editorial line |
language of publication |
individuals/community involved in the event |
media owner (public/private) |
|
print corpus |
National, regional |
|
French, English |
Individuals and community |
Public and private |
|
Online corpus |
|
|
French, English |
Individuals and community |
Private |
Source: The author
- On the other hand, the main amendment of this article should be the inclusion, as part of the conclusions, melt within the results or in an individual section (5), a discussion. The article needs to deep into the intersection between the theoretical framework and the descriptive results it shows.
To address this concern, I have rewritten a more elaborated conclusion. The subsequent change is as follows:
- Old conclusion
5. Conclusion
In sum, the analysis of the designation of events and actors in newspapers and online publications in our corpus around the fight against terrorism in Cameroon reveals changes in the dynamics of the media sphere (censorship, government repression and economic pressures etc.) Cameroon. In fact, with the establishment of social networks as the new place for information in Cameroon, there has been a transformation in the designation of events and actors in the fight against Boko Haram terrorism, given the ethnic identity issues at stake in the socio-political environment. The quality of the designation observed in the forums studied reflects a rise in hate speech in the ethno-political game in Cameroon. This phenomenon remained controllable with the action of state regulatory bodies. Unfortunately, the flexibility of online content combined with the absence of state control have given free rein to the rise of inter-ethnic discursive hatred in politics, according to our analytical data.
- New Conclusion
5. Conclusion
In summary, the advent of social media in the dynamics of the Cameroonian media landscape has led to significant changes in journalism practices. It has spurred the emergence of new digital actors who now share the power to interpret news with journalists. This study observed this phenomenon through the analysis of event and actor designations in both print and online publications in our corpus, within the context of the fight against terrorism in Cameroon.
The findings indicate that the rise of social media as new information platforms in Cameroon has transformed the designation of events and actors in the fight against Boko Haram terrorism, intertwining it with the ethnic identity issues of the sociopolitical environment. While this reality was already present in print media, its migration to online content has metastasized into hate speech. This phenomenon, which was once controllable through the actions of state regulatory bodies, now reflects a transition from a press discourse environment regulated by state institutions to a realm where the state has limited control means. Consequently, full censorship and other forms of government repression are challenging to enforce.
Describing the relationship between the media and the state under the old regime (1960-1982) in Cameroon, Thomas Athenga (2004) speaks of an environment where journalists had to primarily be patriots or even development agents who unquestionably adhered to the President's discourse. Journalists in Cameroon during this period were "conveyors of propaganda, sedative, calming, and soporific information in the name of the imperatives of nation-building, maintenance of public order, and the preservation of good morals. They were dedicated proselytizers, brokers of the 'father of the nation's' speech and the activities of his administration" (Atenga, 2004: 38). In other words, journalists' work essentially conformed to a "grammar of flattery" (Atenga, 2009: 72), where connivance prevailed, blending marketing, propaganda, and political advertising.
With the arrival of Paul Biya's new regime in 1982 and the liberalization of the press and associative movement in 1990, the Cameroonian state must now learn to deal with a press that can criticize and express opinions on certain decisions. Since the liberalization of the communication sector in the 1990s, establishing a newspaper in Cameroon requires very few administrative procedures.
This did not happen spontaneously. It first involved a nearly five-year struggle to transition from the authorization regime to the declaration regime, which was formalized in January 1996 through the amendment of the 1990 law endorsing freedom of communication. Until then, newspapers still faced the issue of prior censorship, a problem that was also abolished with the amendment of the same law. In this context, the state and the media maintain conflictual, even macabre, relations (Atenga, 2004), within a sociopolitical environment described as authoritarian. Given the state apparatus's limited tolerance for criticism, a new era has emerged, witnessing an intensification of judicial repression against the media and journalists. Over 300 press trials were recorded in the early 1990s-2000s (Atenga, 2004: 5). "The cumulative infringements on freedom of expression (prior censorship, seizures, suspensions, bans, raids on editorial offices, physical assaults, imprisonment of journalists, fiscal harassment, and various forms of intimidation) have far exceeded 2000" (Atenga, 2004: 5). "The announcement of a presidential visit abroad or the Chief's vacation destination can lead to a trial for endangering state security, while in democracies, the president's agenda and movements, as well as those of ministers and other personalities, are made public days in advance… A journalist can be condemned by a court for the substitution or possession of administrative documents because he dared to present the judge with evidence of his allegations. The penalty is even more severe when he does not have them. In any case, he goes to trial already condemned" (Atenga, 2004: 10). This helps understand the limited maneuvering room in the use of event and actor designations.
However, with the advent of Web 3.0 in journalism, a new type of information relationship is now dominant through digital social networks (DSN). This has resulted in the infiltration of hate speech into the political-identity game in Cameroon. Our research results have sufficiently highlighted this through the uses of actor and event designations by web users. These designations are articulated with political-identity ethnic issues, demonstrating that media designation in Cameroon is a site of power relations.
The theoretical foundations of the relationship between identity and media are rooted in the writings of Hall (2007: 203-214), particularly in "New Ethnicity." According to Hall, the relationship between identity and media's discursive strategy is based on an ideological struggle, raising the question of "representation relationship" (Ibid: 204). Based on these approaches, the media shape our understanding of ethnicity, suggesting how it should be perceived, understood, and the meanings it conveys. This practice relies on significant effort in media representation. Hall emphasizes that "representation is only possible because enunciation always occurs in codes that have a history, occupying a position in the discursive forms of a specific time and space" (Hall, 2007: 209). In other words, "we all speak from a place, a particular experience, a specific history, a particular culture. [...] And in this sense, we are all ethnically located, and our ethnic identities are crucial to the subjective sense of who we are" (Ibid, 210).
In the Cameroonian context, this perspective views ethno-regional identities as a source of power in the realm of political competition. In other words, belonging to a specific ethno-regional group would be a political asset. This identity-centric vision of political power is responsible for the transposition of the reading of ethnic identity into the Cameroonian media field. Due to the stakes related to political competition, Cameroonian newspapers are labeled based on ethno-regional identities. Therefore, "the process of determining events and situations does not solely rely on the media and involves a plurality of actors" (Arquembourg, 2005: 29).
This approach refers to media designations as a mediation site among various actors, stakeholders in a social reality (Garric, 2009). In Ricœur's terminology (1983), these are precisely narrative operations in the mimetic process, allowing an understanding of how informative narratives achieve their objectives (Arquembourg, 2005).
Therefore, we conceptualize designations as sites of incorporation (Granovetter, 1985; D'Hont, Gérard, 2015, Le Velly, 2002; Laville, 2008, etc.) of power relations in society. The representational approach of the media often results from the intertwining or entanglement of various power relations operating in society (Bobo, 1995; Hill Collins, 1990: 68-69; Hooks Bell 1992, 1996, Hall, 2007). These can be identity-based, religious, cultural, racial, and often economic. Therefore, Le Velly (2002) apprehends incorporation as a sociology of market exchanges at the heart of social construction processes. Applied to the media domain, the logic of incorporation (Granovetter, 1985; D'Hont, Gérard, 2015, Le Velly, 2002; Laville, 2008, etc.) accounts for the fact that the interests of media actors and those of economic and political environments form a system. This would result in event coverage based on the agreement model (Boltanski, Thévenot, 1991; Legavre, 2011).
In the Cameroonian sociopolitical context influenced by the philosophy of communitarian liberalism (2018, 1986), this situation reveals various layers/surfaces of incorporation and power relations that intersect. Particularly, ethnic power relations, often associated with language, region, tribe, etc., further complicate things from an analytical perspective. It becomes challenging to ultimately identify what guides the editorial orientation of newspapers, including online content. Our perspective also considers designations as editorial processes crystallizing current events within frames of ethnic and territorial identity in a sociopolitical environment governed by competition on power issues. In other words, concerning the news studied in this research, designations of actors and events are primarily means of ethnic identification and territorialization in newspapers and online forums. To put it differently, the observed media designations in our corpus are also sites of ethno-political territorialization of the Cameroonian national space. Indeed, the territorialization of the Cameroonian national space is strongly oriented towards the identity diversity (ethnic and linguistic) characterizing the country.
This initiative is part of the state's desire for "territorial differentiation" (Isabelle Pailliart, 2013: 278). It is a long-term process, initiated by both the state and the media, aiming not only to designate the multiplication of territorial levels marking the country's map but also, more fundamentally, the movement of differentiation in the social world (Pailliart, 2013). In Cameroon, the media heavily relies on the state's policy of territorialization. There are national, regional, departmental, etc., media entities that contribute to mastering territorial differentiation and, by extension, social differentiation. It is in this aspect that we consider the territoriality of the newspaper in the analysis of the designation we conducted. That is, as a movement of differentiation of ethno-regional identity. An approach now strongly contested with the advent of social networks, whose territoriality is global. Thus, they challenge traditional obstacles such as censorship, government repression, and economic pressures, etc.
The way in which Social Media Networks (SMN) permeate the news studied in this research ultimately highlights the issue of media regulation in a multi-ethnic context in Africa. Ethnicity is a crucial variable around which power struggles are organized. Therefore, how should media regulation be reorganized in Cameroon in view of this challenge posed by SMN? Indeed, in addition to SMN, Cameroon is entrenched in a context of the end of a reign of over 40 years at the helm of the state, and where the press unfolds a "tribal war through intermediary newspapers [where] particularisms are thus magnified and defended. These are deviations that are justified, but when taken too far, can prove very dangerous. No longer are ideas or ideologies defended. One defends the tribe through the political party and through the loyal press" (Ndembiyembe, 1997: 47). We believe that a reorganization of media regulation in Cameroon, considering the results of our study, should more closely align with local socio-political realities. The idea is to reconsider ethical and deontological standards specific to the Cameroonian media environment.
Round 2
Reviewer 2 Report
Comments and Suggestions for Authors
The author has addressed most of the comments I expose in the review. The article flows better in this new version.
Author Response
Thank you for your observations, which I receive with humility. Below are the corrections requested
I acknowledge the feedback received regarding the author's narrative style, particularly in the application of Critical Discourse Analysis. It has been noted that explicit clarity or demarcation of research questions is not strictly required, as these questions are interwoven within the narrative of the relevant sections. However, to enhance overall clarity, it is advisable to:
a. On pages 2-3, present the research questions (RQs) initially, separate from the accompanying explanatory narrative (94, 101, 102, 103, 104...). The elucidation should then follow the articulation of the RQs.
Following this observation, I have ensured that the research questions appear at the beginning of page 3.
b. Additional RQs were posed in relation to the discussion of ‘Designation (170, 171, 172, 173…) Are these central RQs as well? Or are they sub questions related to your initial RQs? My sense is that they are additional RQs.
Thank you for this observation. I did not initially conceptualize these questions as subordinate sub-questions. Thus, I have reformulated this section of my text. Previously, it was written as follows: "Therefore, how does designation operate in the newspaper articles and online content within our corpus? In what way does it account for ethnic identity dynamics?" Here is the refined version:
"In the subsequent sections, we will investigate the functioning of designation in newspaper articles and online content within our corpus. Additionally, we will scrutinize its implications for dynamics associated with ethnic identity."
c.Similarly, the author has additional RQs in discussion of ‘Event – Actor Designation’ (263, 264, 265.
Upon re-examination, unless I am mistaken, I did not find a question at the specified lines or within the subsection.
we need to know how the specific RQs posed by the study were answered or not answered.
In other words, clarity would be enhanced in the conclusion and discussion through a focused exploration of the primary research inquiry. However, as previously indicated, the lines subject to questioning, particularly in the discussion of 'Designation' (170, 171, 172, 173...), have been reformulated, eliminating the presence of secondary questions.
